# Absence of Grail promotes CD8[+] T cell anti-tumour activity

Cara Haymaker [1], Yi Yang[2,3], Junmei Wang[2], Qiang Zou[2], Anupama Sahoo[2], Andrei Alekseev[2], Divyendu Singh[2], Krit Ritthipichai[1], Yared Hailemichael[1], Oanh N. Hoang[2], Hong Qin[4,5], Kimberly S. Schluns[2], Tiejun Wang[3], Willem W. Overwijk[1], Shao-Cong Sun[2], Chantale Bernatchez[1], Larry W. Kwak[4,5], Sattva S. Neelapu[4] & Roza Nurieva[2]

T-cell tolerance is a major obstacle to successful cancer immunotherapy; thus, developing strategies to break immune tolerance is a high priority. Here we show that expression of the E3 ubiquitin ligase Grail is upregulated in CD8[+] T cells that have infiltrated into transplanted lymphoma tumours, and Grail deficiency confers long-term tumour control. Importantly, therapeutic transfer of Grail-deficient CD8[+] T cells is sufficient to repress established tumours. Mechanistically, loss of Grail enhances anti-tumour reactivity and functionality of CD8[+] T cells. In addition, Grail-deficient CD8[+] T cells have increased IL-21 receptor (IL-21R) expression and hyperresponsiveness to IL-21 signalling as Grail promotes IL-21R ubiquitination and degradation. Moreover, CD8[+] T cells isolated from lymphoma patients express higher levels of Grail and lower levels of IL-21R, compared with CD8[+] T cells from normal donors. Our data demonstrate that Grail is a crucial factor controlling CD8[+] T-cell function and is a potential target to improve cytotoxic T-cell activity.

[1] Department of Melanoma Medical Oncology, M.D. Anderson Cancer Center, 1515 Holcombe Blvd., Houston, TX 77030, USA. [2] Department of Immunology, M.D. Anderson Cancer Center, 1515 Holcombe Blvd., Houston, TX 77030, USA. [3] Department of Radiation Oncology, The Second Hospital of Jilin University, No. 218 Ziqiang St., Changchun City, Jilin Province 130041, China. [4] Department of Lymphoma/Myeloma, M.D. Anderson Cancer Center, 1515 Holcombe Blvd., Houston, TX 77030, USA. [5] Toni Stephenson Lymphoma Center, City of Hope, 1500 East Duarte Rd., Duarte, CA 91010, USA. Cara Haymaker and Yi Yang contributed equally to this work. Correspondence and requests for materials should be addressed to R.N. (email: rnurieva@mdanderson.org)

The adaptive immune system, especially CD8[+] cytotoxic T lymphocytes (CTL), have a crucial function in controlling the development of neoplastic lesions[1]. Effector CD8[+] T cells can efficiently destroy target cells with death cell ligands such as tumour necrosis factor (TNF)-related apoptosis-inducing ligand (TRAIL) or execution of the perforin/granzyme and interferon (IFN)-γ-dependent machinery[2]. Although CD4[+] T cells are important in anti-cancer immunity, their predominant function at tumour sites is to maintain function of tumour-specific CTLs by producing cytokines[3, 4].

Cancer immunotherapy aims to reactivate a patient's immune system to fight against tumours; however, T-cell tolerance induced by the inhibitory tumour microenvironment is an obstacle[5]. Therefore, understanding the cellular and molecular mechanisms that underlie T-cell tolerance in cancer would guide the development of effective therapies. E3 ubiquitin ligases,

**Fig. 1** EG-7 lymphoma growth is dramatically reduced in the absence of Grail. **a** WT and *Grail*[−/−] mice were injected with EG-7 tumour cells subcutaneously on day 0. Tumour growth was monitored from day 7 and calculated as ½(*ab*[2]), where *a* is the length and *b* is the width. EG-7 tumour weight in WT and *Grail*[−/−] mice after euthanasia is shown on the *right graph*. (*n* = 3 independent experiments with a total of 18 WT mice and 15 *Grail*[−/−] mice). **b** Tumour-infiltrating lymphocytes isolated from EG-7 tumours on days 7 and 12 were stained with anti-CD4 and CD8α antibodies. Numbers in dot plot indicate the percentage of cells within each quadrant from a representative tumour at each time point. Individual mice per group are shown for each time point on the *right bar graph*. (*n* = 10 mice per group from two experiments). **c** TIL were isolated from EG-7 tumours on day 17 and stained for CD4[+] and CD8[+] T cells as in **b**. The *right bar graph* shows the percentage of CD4[+] and CD8[+] T-cell subsets from individual mice per group. (*n* = 18 WT mice and 15 *Grail*[−/−] mice from three independent experiments). **d** TILs were stained with an OVA-specific pentamer and analysed in the CD8[+] gate. (*n* = 18 WT mice and 15 *Grail*[−/−] mice from three independent experiments). **e** TILs were stimulated for 5 h with OVA peptide, followed by intracellular staining of IFN-γ and granzyme B (GzmB) and analysed in the CD8[+] gate. Numbers in dot plot quadrants represent the percentages of each marker. The percent expression of CD8[+]IFNγ[+]GzmB[+] TILs per mouse is shown on the *right bar graph*. (*n* = 18 WT mice and 15 *Grail*[−/−] mice from three independent experiments) *$p < 0.05$, **$p < 0.01$, ****$p < 0.0001$, NS as determined using one-way ANOVA. *NS* not significant

including Cbl-b, Itch and Grail are important regulators of T-cell tolerance[6]. Cbl-b and Itch have been reported to be involved in tumour development[7–10] and expression of Grail, a type I transmembrane protein localised to the endosomal compartment, is associated with T-cell anergy[11]. We previously showed that Grail-deficient mice are resistant to immune tolerance induction in vitro and in vivo[12, 13]. We showed that Grail is required for downregulating TCR signalling in recently activated CD4$^+$ T cells, and that lack of Grail leads to hyperproliferation, excessive cytokine production and abrogation of the suppressive function of regulatory T (Treg) cells[12]. However, the role of Grail in CD8$^+$ T cells is unclear.

In the current study, we find high expression of Grail in mouse CD8$^+$ T cells that have infiltrated into lymphoma tumours and we examine the role of Grail in EL-4 and EG-7 lymphoma models. Grail deficiency provides the host with spontaneous protection against tumours, which is mediated mainly by CD8$^+$ T cells in Grail-deficient mice. In tumours, loss of Grail enhances anti-tumour reactivity of CD8$^+$ T cells. Moreover, in mouse CD8$^+$ T cells, Grail regulates the expression of IL-21 receptor (IL-21R) and naive Grail$^{-/-}$ CD8$^+$ T cells are hyperresponsiveness to IL-21 in vitro due to augmented IL-21R expression, which is regulated by Grail-dependent ubiquitination and degradation. Importantly, similar to the mouse model, we show high expression of Grail and diminished expression of IL-21R in CD8$^+$ T cells from lymphoma patients compared with healthy donors. Altogether, our results indicate a suppressive function of Grail in CD8$^+$ T cells and suggest that Grail-ablated CD8$^+$ T cells could be an efficient tool for eliciting immune responses against tumours.

## Results

**Grail loss correlates with enhanced CD8$^+$ TILs infiltration.** To investigate the potential role of Grail in CD8$^+$ T cells, we utilised subcutaneous EL-4 and EG-7 murine lymphoma models. EL-4 cells are derived from a T-lineage lymphoma developed in C57BL/6 mouse treated with 9,10-dimethyl-1,2-benzanthracene and represents a tumour model with weak immunogenicity. EG-7 cells are EL-4 transfectants that express chicken ovalbumin (OVA), which serves as tumour antigen and are therefore considered highly immunogenic tumours. EG-7 lymphoma cells were injected subcutaneously into the flanks of wild-type (WT) mice and 17 days later CD4$^+$ and CD8$^+$ tumour-infiltrating lymphocytes (TIL) or T cells residing in the spleen were sorted and analysed for Grail expression by real-time (RT)-PCR. Remarkably, Grail messenger RNA (mRNA) levels were significantly upregulated in CD8$^+$ T cells from tumours compared to those in spleens, suggesting a role of Grail in controlling the function of tumour-specific CTLs (Supplementary Fig. 1a). In contrast, we did not detect any significant upregulation of Grail in tumour-infiltrated CD4$^+$ T cells. Interestingly, Cbl-b expression was not increased in CD4$^+$ and CD8$^+$ TILs (Supplementary Fig. 1b), suggesting a distinct regulation and function of Grail and Cbl-b in tumours. Similarly, when EL-4 cells were injected in WT mice, Grail but not Cbl-b expression was selectively upregulated in CD8$^+$ T cells infiltrated in tumours (Supplementary Fig. 1c and d), suggesting that both strong or weak immunogenic tumours selectively induced Grail expression in tumour-infiltrating CD8$^+$ T cells in vivo.

To assess whether Grail contributes to anti-tumour immunity in vivo, we inoculated EG-7 cells into sex- and age-matched WT and Grail$^{-/-}$ mice and monitored the kinetics of tumour growth (Fig. 1a). Tumour growth was comparable between WT and Grail$^{-/-}$ mice in the first 10 days after tumour inoculation. Intriguingly, while between day 10 and 17 tumours grew rapidly in WT mice; Grail-deficient mice were able to control tumour growth until day

55 demonstrating a delay in tumour growth in the absence of Grail (Fig. 1a; Supplementary Fig. 2a). To explore the underlying immune mechanisms of reduced tumour growth in Grail$^{-/-}$ mice, we analysed TILs. In comparison to TILs from WT mice, we observed that the percentage of total CD8$^+$ TILs is comparable at day 7 but increased by day 12 in Grail$^{-/-}$ mice (Fig. 1b). This increased frequency of tumour antigen-specific CD8$^+$ TILs from Grail$^{-/-}$ mice was maintained at day 17 (Fig. 1c, d); however, we did not detect changes in the percentage of Grail$^{-/-}$ CD4$^+$ TILs compared to WT (Fig. 1b, c). By day 55, the ratio of CD4$^+$:CD8$^+$ TIL was 1:1 but CD8$^+$ TIL were still found to have cytotoxic capability as demonstrated by granzyme B and IFNγ expression (Supplementary Fig. 2b). Similarly to the EG-7 model, EL-4 tumours rapidly grew in WT mice, while being well-controlled in Grail$^{-/-}$ mice (Supplementary Fig. 3). Moreover, the total number of CD8$^+$ T cells but not CD4$^+$ T cells was increased in the tumours of Grail$^{-/-}$ mice inoculated with EL-4 tumour cells compared to WT mice. Thus, these data show that loss of Grail expression results in increased infiltration of CD8$^+$ T cells into the tumour tissues and importantly in attenuation of tumour growth with either strong or weak immunogenicity.

For further studies, we evaluated the accumulation, activation and effector function of tumour antigen-specific CD8$^+$ T cells. First, we assessed whether accumulation of Grail$^{-/-}$ CD8$^+$ TILs could be due to either a decreased suppressive capability of Grail$^{-/-}$ Tregs or a gained resistance to suppression by the CD8$^+$ T cells. To address this possibility, we examined the suppressive function of Treg cells from Foxp3-GFP and Grail$^{-/-}$ Foxp3-GFP mice towards WT and Grail$^{-/-}$ CD8$^+$ T cells (Supplementary Fig. 4a). Similar to CD4$^+$ T cell, CD4$^+$ Foxp3-GFP$^+$ T cells from Grail$^{-/-}$ mice exhibited reduced suppression activity to both WT and Grail$^{-/-}$ CD8$^+$ T cells[12]. However, Grail$^{-/-}$ CD8$^+$ T cells appeared to have gained a resistance to WT Treg-mediated suppression both in terms of proliferation and production of granzyme B and IFNγ (Supplementary Fig. 4a). This suggests that the delayed tumour growth observed in vivo may be due to a combination of reduced suppressive activity of Tregs and an enhanced resistance to suppression by CD8$^+$ T cells in the absence of Grail.

To further explore the possible mechanism(s) for accumulation of Grail$^{-/-}$ CD8$^+$ T cells in the EG-7 model of lymphoma, next we analysed the susceptibility of WT and Grail$^{-/-}$ CD8$^+$ TILs to apoptosis; however, we did not detect any difference in the percentage of apoptotic markers Annexin V and 7AAD in CD8$^+$ TIL between WT and Grail$^{-/-}$ groups on days 7 and 12 after tumour inoculation. In addition, we analysed the expression of pro-survival costimulatory receptors OX-40 and 4-1BB and anti-apoptotic factor Bcl-2 in CD8$^+$ TIL from WT and Grail$^{-/-}$ mice. Interestingly, CD8$^+$ TIL from Grail$^{-/-}$ mice had increased expression of OX-40, 4-1BB and Bcl-2 at both time points (Supplementary Fig. 4b). Moreover, mRNA levels of pro-survival costimulatory molecules OX-40 and 4-1BB, and anti-apoptotic gene Bcl-xL (B-cell lymphoma-extra large) were enhanced on day 17 as well, suggesting that Grail could control the survival of CTLs infiltrated into the tumours and thus result in the accumulation of this subset within the tumour (Supplementary Fig. 4c).

To assess whether the absence of Grail has an effect on activation and effector function of CD8$^+$ T cells, we analysed OVA-specific responses ex vivo upon OT-I peptide (chicken OVA peptide 257–264 SIINFEKL) restimulation. In fact, we observed a significantly higher percentage of IFN-γ and granzyme B double-positive cells in Grail$^{-/-}$ CD8$^+$ T cells compared to WT CD8$^+$ T cells (Fig. 1e), suggesting that Grail ablation elicits effector function of CTLs. Since WT CD8$^+$ T cells also highly expressed granzyme B, the impact of Grail loss may be specific to

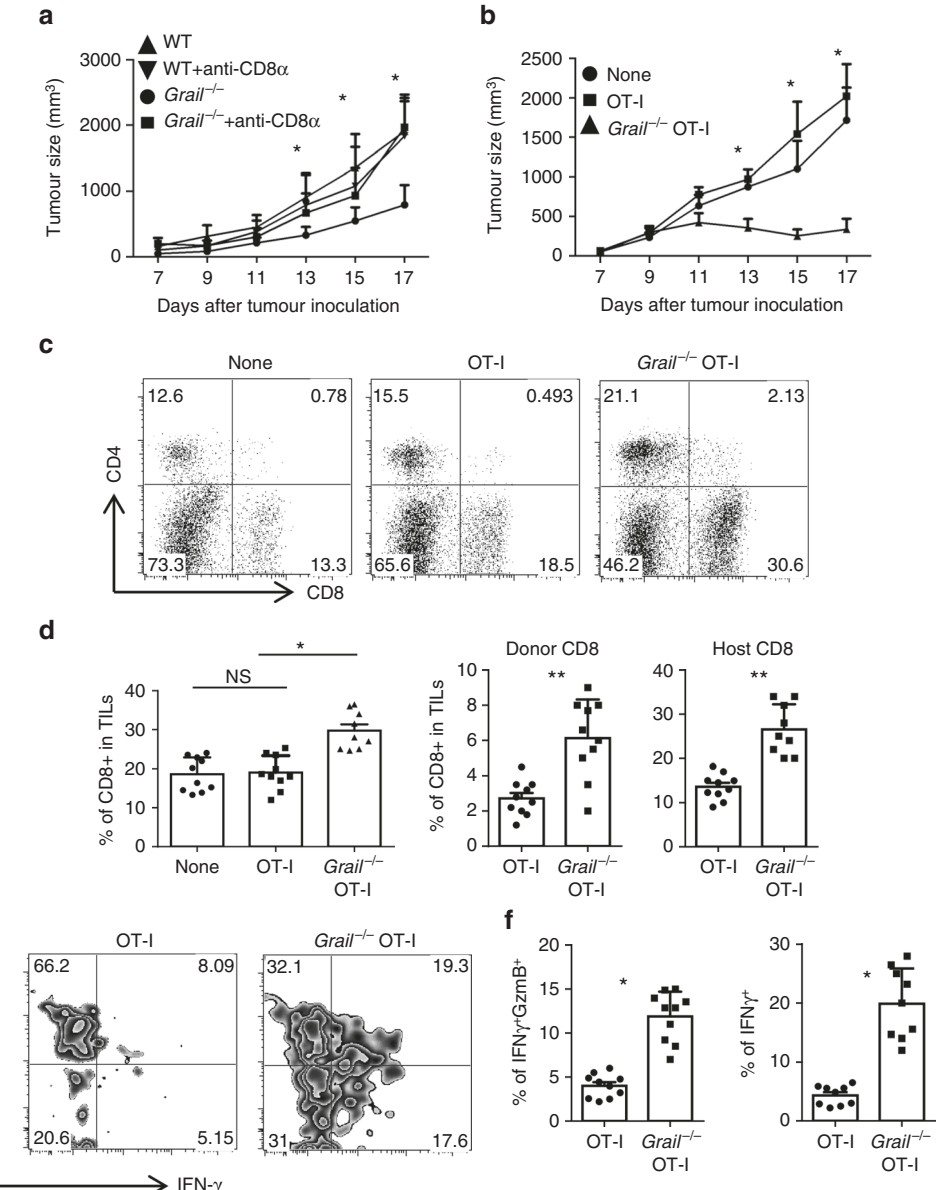

**Fig. 2** *Grail*[−/−] CD8[+] T cells are necessary and sufficient for tumour control. **a** WT and *Grail*[−/−] mice were injected with rIgG or anti-CD8α antibody intraperitoneally 3 days before EG-7 tumour cell inoculation and weekly after that throughout the duration of the experiment. Tumour size was measured and calculated as in Fig. 1. (*n* = 2 independent experiments with 5–7 mice per group) **b–e** CD45.1[+] congenic mice were inoculated with EG-7 tumours and on day 5 (when tumours are palpable), were injected i.v. with purified CD45.2[+] OT-I or *Grail*[−/−] OT-I cells. A group of mice that did not receive OT-I cells was used as control (None). **b** Tumour size in all recipient mice was calculated as in Fig. 1. (*n* = a total of 10 mice per group from two experiments). **c, d** The percentage of total CD8[+] T cells and CD4[+] T cells in TILs were analysed. A representative dot plot is shown in **c**. **d** The percentage of CD8[+] TIL from each group is shown from individual mice on the *left bar graph*. The *middle bar graph* shows the percentage of donor CD45.2[+]CD8[+] TIL and the *right bar graph* shows the percentage of host CD45.1[+]CD8[+] TIL from each mouse (*n* = 10 mice per group from two experiments). **e, f** TILs were stimulated for 5 h with OVA peptide and the production of IFN-γ and granzyme B (GzmB) was analysed in the CD45.2[+]CD8[+] gate by intracellular staining. A representative zebra plot is shown in **e**. **f** The *bar graph* shows the percentage of IFNγ[+]GzmB[+] and IFNγ[+] subsets per mouse (*n* = 10 mice per group from two experiments). **d, f** Results are shown as mean ± SEM. *$p < 0.05$, **$p < 0.01$, NS as determined using a Student's *t*-test. *NS* not significant

IFNγ responses. As we did not detect any IFNγ single producing cells, we focused on the granzyme B[+]IFNγ[+] subset.

To confirm the functional state of *Grail*[−/−] CD8[+] T cells in the tumours, we sorted tumour-infiltrating CD8[+] T cells from WT and *Grail*[−/−] mice and analysed the expression of cytolytic factors by RT-PCR. We detected a statistically significant increase in mRNA levels of IFN-γ, granzyme B, TNF-α and perforin 1 in CD8[+] TIL from *Grail*[−/−] mice (Supplementary Fig. 4c), further supporting that Grail controls the effector function of CTLs infiltrated into the tumours.

***Grail*[−/−] CD8[+] T cells are essential for EG-7 tumour control.** To test whether CD8[+] T cells were necessary to control tumour growth in *Grail*[−/−] mice, we depleted CD8[+] T cells using a specific antibody before and during EG-7 tumour growth. For WT mice, both CD8[+] T cell-depleted and rIgG-treated groups showed similar tumour growth (Fig. 2a), suggesting the nonresponsive state of WT CD8[+] T cells. Interestingly, tumour growth was dramatically increased when CD8[+] T cells were depleted in *Grail*[−/−] mice (Fig. 2a), supporting the critical role of CD8[+] T cells in tumour control in *Grail*[−/−] mice.

Next we examined whether absence of Grail in CD8+ T cells would be sufficient to confer a protective role against established tumours in normal host. To answer this, we used an adoptive cell transfer therapeutic model where *Grail*−/− mice were crossed with OT-I TCR transgenic mice to obtain *Grail*−/− OT-I mice. CD8+ T cells (CD45.2+) from OT-I and *Grail*−/− OT-I mice were transferred intravenously into congenic CD45.1 mice 5 days after EG-7 inoculation when tumour was palpable. The kinetics and extent of tumour growth were similar between WT control mice (no cell transfer) and mice that received WT OT-I cells (Fig. 2b).

In contrast, tumour growth was markedly reduced in mice which received *Grail*−/− OT-I cells (Fig. 2b). These data indicated that therapeutic transfer of OVA-specific *Grail*−/− CD8+ T cells was sufficient to mediate regression of established EG-7 tumours.

To investigate the underlying mechanisms, we analysed the transferred CD8+ T cells in TILs. We found that there was a marked increase in percentage of total, donor and host CD8+ T cells in TILs from the *Grail*−/− OT-I-transferred group (Fig. 2c, d), while there was no difference between control group and OT-I-transferred group. The increase in host CD8+ T cells could be

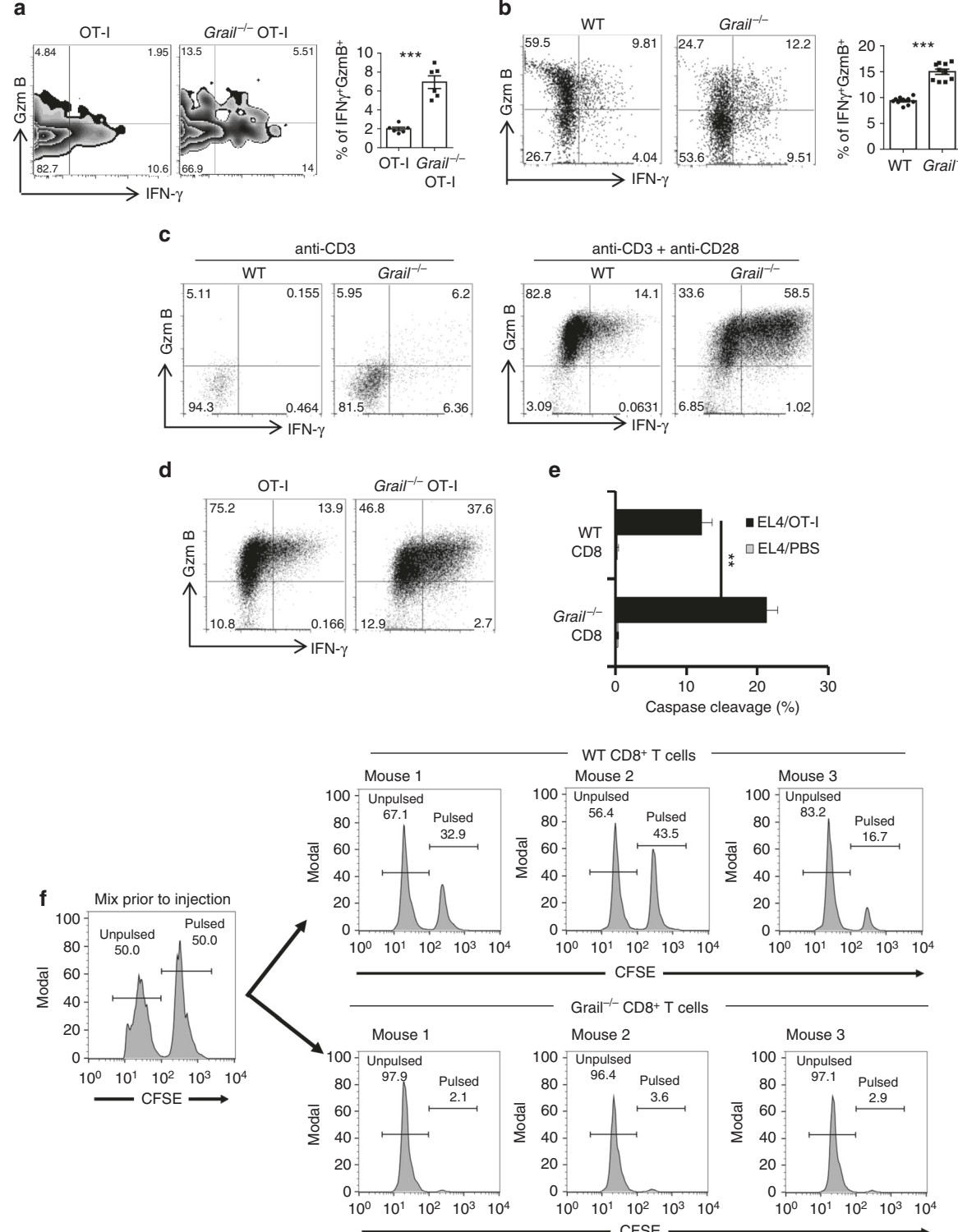

due to the increased level of apoptosis and/or inflammation at the tumour site mediated by $Grail^{-/-}$ $CD8^+$ T cells. Analysis of transferred OVA-specific $CD8^+$ T cells showed significantly higher percentage for IFN-γ and granzyme B double-positive or total IFN-$γ^+$ cells in transferred $Grail^{-/-}$ OT-I cells than WT OT-I cells (Fig. 2e, f). These data suggested that in the therapeutic model, transferred $Grail^{-/-}$ OT-I cells retained their cytolytic function in the suppressive tumour microenvironment and are sufficient to suppress established tumours.

**Enhanced effector function of Grail-deficient $CD8^+$ T cells**. In order to understand the mechanism for enhanced function of Grail-deficient $CD8^+$ T cells, first we analysed the role of Grail during $CD8^+$ T-cell activation in vivo. Purified $CD45.2^+$ OT-I and $Grail^{-/-}$ OT-I cells were labelled with 5-(and 6)-Carboxyfluorescein diacetate succinimidyl ester (CFDA SE, CFSE) and adoptively transferred into $CD45.1^+$ congenic mice that were inoculated with EG-7 tumour cells 5 days earlier. Three days later, proliferation of transferred cells and their cytokine production were examined. We found that proliferation and IL-2 production by transferred $Grail^{-/-}$ OT-I cells were comparable to WT OT-I cells (Supplementary Fig. 5a, b). However, in contrast to WT OT-I cells, $Grail^{-/-}$ OT-I cells in TILs showed a higher frequency of IFN-$γ^+$ granzyme $B^+$ expression upon in vitro restimulation with OT-I peptide (Fig. 3a), which suggested that Grail negatively regulates $CD8^+$ T-cell effector function in vivo.

To check whether the increased $CD8^+$ T-cell function in Grail-deficient mice is T-cell intrinsic, we generated mixed bone marrow chimeras by transferring a mixture of congenic $CD45.1^+$ WT and $CD45.2^+$ Grail-deficient bone marrow cells into sublethally irradiated $Rag1^{-/-}$ mice. Eight weeks after reconstitution, we inoculated mice with EG-7 tumour cells. Grail-deficient $CD8^+$ T cells infiltrated into the tumours exhibited greatly enhanced production of IFN-γ when compared to WT cells in response to OT-I restimulation (Fig. 3b). The above results confirm that Grail expression in T cells is required to control $CD8^+$ T-cell effector function in vivo.

To further analyse the function of Grail in $CD8^+$ T cells, naive $CD8^+$ T cells from WT and $Grail^{-/-}$ mice were activated with plate-bound anti-CD3 alone or together with anti-CD28. WT and $Grail^{-/-}$ $CD8^+$ T cells exhibited similar proliferation and IL-2 production (Supplementary Fig. 5c). Consistent with the above results, $Grail^{-/-}$ OT-I cells showed comparable proliferation and IL-2 production with WT OT-I cells when activated with OVA peptide and WT antigen-presenting cells (APCs) (Supplementary Fig. 5d). These data indicate that in contrast to $CD4^+$ T cells, Grail does not control proliferation and IL-2 production in $CD8^+$ T cells[12]. We also activated OT-I or $Grail^{-/-}$ OT-I splenocytes

using anti-CD3 or OVA peptide and checked the upregulation of activation markers CD69, CD44 and CD25 on $CD8^+$ T cells. There was no difference in upregulation of these markers between WT and $Grail^{-/-}$ $CD8^+$ T cells (Supplementary Fig. 6). However, we found that $Grail^{-/-}$ $CD8^+$ T cells exhibited greatly increased production of IFN-γ in response to anti-CD3 stimulation alone, which was further enhanced by addition of anti-CD28 (Fig. 3c). Similarly, significant differences in IFN-γ production between WT and $Grail^{-/-}$ OT-I cells were observed in the presence of OVA peptide (Fig. 3d). Most importantly, significant improvement in cytokine expression in $Grail^{-/-}$ $CD8^+$ T cells led to their enhanced killing function both in vitro and in vivo compared to WT $CD8^+$ T-cell effectors (Fig. 3e, f). An extended in vivo killing assay demonstrates that WT $CD8^+$ T cells required additional time to perform their efficient antigen-specific cytolytic function as compared to $Grail^{-/-}$ $CD8^+$ T cells (Fig. 3f; Supplementary Fig. 7). These results indicate that Grail controls the cytotoxic function of $CD8^+$ T cells.

Previously we reported that Grail-deficient $CD4^+$ T cells are less efficient in downregulation of TCR-CD3 expression[12]. To assess whether enhanced cytotoxic function of $Grail^{-/-}$ $CD8^+$ T cells is associated with their inefficiency in TCR downmodulation, WT and $Grail^{-/-}$ OT-I T cells were activated with OVA peptide and changes in TCRβ expression analysed by flow cytometry over a 6 h time course. OVA peptide stimulation led to TCR downmodulation on WT cells, whereas TCR modulation was markedly attenuated in $Grail^{-/-}$ $CD8^+$ T cells, suggesting that Grail-deficient $CD8^+$ T cells are more resistant to activation-induced TCR downmodulation than WT cells (Supplementary Fig. 8a). To investigate further, we analysed the ubiquitination of CD3ζ in $CD8^+$ T cells from WT and $Grail^{-/-}$ mice. Six hours after activation in the presence of MG132 to prevent degradation, T-cell lysates were subject to immunoprecipitation with CD3ζ antibodies and immunoblotting with anti-Ub antibodies. In comparison to WT cells, ubiquitination of CD3ζ in $Grail^{-/-}$ $CD8^+$ T cells was substantially reduced demonstrating that Grail targets CD3 through its ubiquitination and subsequent degradation (Supplementary Fig. 8b). Thus, sustained TCR-CD3 signalling in $Grail^{-/-}$ $CD8^+$ T cells could be one of the possible mechanisms for their elevated effector function.

**Grail loss causes hyperresponsiveness to IL-21R signalling**. Members of the common γ-chain (γc) cytokine family, including IL-2, IL-7 and IL-15 have important roles in regulating $CD8^+$ T-cell response. IL-21, a member of the γc family, has also been reported to improve $CD8^+$ T-cell function through regulation of IFN-γ, granzyme B and perforin production, and survival in infection and tumour models[14–16]. Interestingly, we found that

---

**Fig. 3** Enhanced effector function of $Grail^{-/-}$ $CD8^+$ T cells in vitro and in vivo. **a** $CD45.1^+$ congenic mice were inoculated with EG-7 tumours and, $CD45.2^+$ OT-I or $Grail^{-/-}$ OT-I cells were transferred 5 days later. Eight days after OT-1 transfer, TILs were isolated and stimulated with OVA peptide stained for IFN-γ and granzyme B (GzmB), and analysed in $CD45.2^+CD8^+$ gate. The *bar graph* shows mean ± SEM as well as the percentage from individual mice per group ($n = 6$ mice from two experiments). **b** Mixed bone marrow chimeric mice were inoculated with EG-7 tumour cells. TILs were stimulated as in **a** and analysed for IFN-γ and GzmB production in $CD45.2^-CD8^+$ (WT) or $CD45.2^+CD8^+$ ($Grail^{-/-}$) gate. The *bar graph* shows mean ± SEM as well as the percentage from individual mice per group ($n = 10$ mice from two experiments). **c** FACS-sorted naive $CD8^+$ T cells were activated with anti-CD3 alone or together with anti-CD28. Three days later, expression of IFN-γ and GzmB was analysed by intracellular staining. $N = 3$ independent experiments. **d** Naive OT-I and $Grail^{-/-}$ OT-I cells were activated with OVA peptide and irradiated WT APCs. Two days later, expression of IFN-γ and GzmB was analysed by intracellular staining. $N = 3$ independent experiments. **e** Two days after activation with OT-I peptide, WT and $Grail^{-/-}$ $CD8^+$ T cells were incubated with 1 µg/ml OVA peptide-pulsed EL-4 cells labelled with DDAO-SE for 2 h. The cells were fixed, permeabilized and stained with anti-cleaved caspase-3 mAb. $N = 3$ independent experiments. **f** Spleen cells from OT-I or $Grail^{-/-}$ OT-I mice were adoptively transferred into C57BL/6J mice ($n = 3$ mice per group) followed by s.c. vaccination with OVA peptide and anti-CD40. Imiquimod cream was applied on the vaccination site. In addition, mice received 100,000 IU rhIL-2 by i.p. Three days after vaccination, mice were injected with a 1:1 mix of target cells loaded with 1 µg/ml peptide and labelled with 5 µmol/l CFSE or unloaded and labelled with 0.5 µmol/l CFSE. Four hours later, splenocytes from the recipients were analysed by flow cytometry **p < 0.01, ***p < 0.001 as determined using a Student's t-test

*Grail⁻/⁻* CD8⁺ TILs expressed significantly higher level of IL-21R compared to WT TILs (Fig. 4a) suggesting a potential role of Grail in the regulation of IL-21 signalling.

To assess whether IL-21R signalling could contribute to enhanced effector function of *Grail⁻/⁻* CD8⁺ T cells, FACS-sorted naive WT and *Grail⁻/⁻* CD8⁺ T cells were activated with

plate-bound anti-CD3 in the absence or presence of IL-21. Interestingly, though WT and *Grail⁻/⁻* CD8⁺ T cells proliferated similarly with anti-CD3 treatment alone, proliferation of *Grail⁻/⁻* CD8⁺ T cells significantly increased after IL-21 treatment compared to WT CD8⁺ T cells (Fig. 4b). In addition, whereas anti-CD3-induced IFN-γ, granzyme B, perforin1 and IL-21Rα

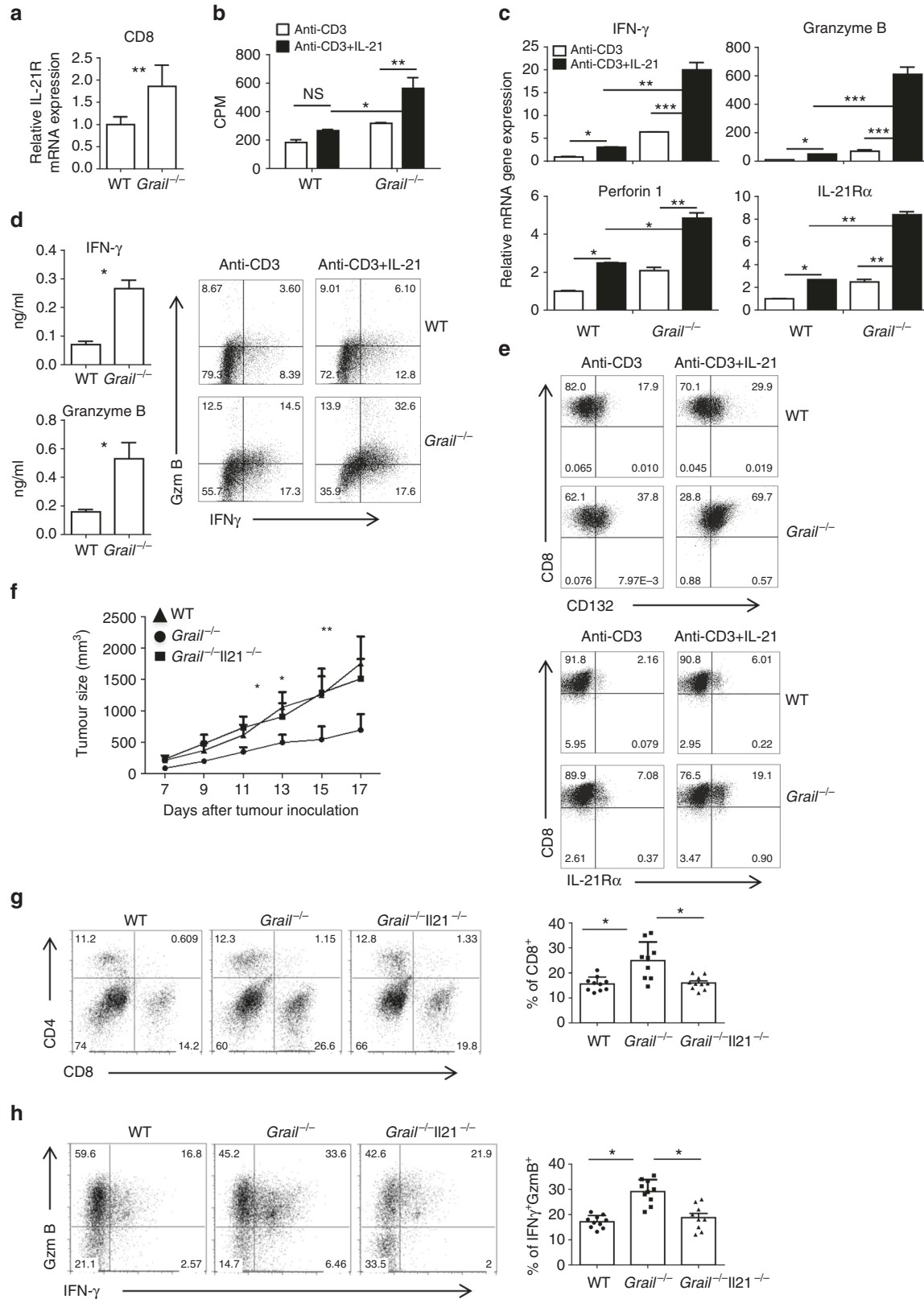

mRNA, and protein expression in WT CD8[+] T cells was increased, there was significantly higher expression of these molecules (both mRNA and protein levels) in Grail[−/−] CD8[+] T cells upon IL-21 treatment (Fig. 4c–e; Supplementary Fig. 9). The enhanced production of cytolytic factors and IL-21R by Grail[−/−] CD8[+] T cells was not simply due to the increased proliferation of these cells in response to IL-21 as this was also observed at the per cell level by flow cytometry staining. These data suggested that as Grail[−/−] CD8[+] T cells express higher levels of IL-21R, they are more sensitive to IL-21 treatment than WT cells and that the expression of IL-21R may be driven in part by IL-21.

To further assess the role of IL-21 signalling in effector function of Grail[−/−] CD8[+] T cells in vivo, WT, Grail[−/−] and Grail[−/−]Il21[−/−] mice were inoculated with EG-7 tumour cells. Interestingly, absence of IL-21 caused progressive tumour growth in Grail[−/−] mice comparable to WT mice (Fig. 4f). Consistent with increased tumour growth, percentage of CD8[+] T cells infiltrated to the tumours of Grail[−/−]Il21[−/−] mice was decreased compared to Grail[−/−] mice (Fig. 4g). However, tumour growth was comparable between WT and Il21[−/−] mice demonstrating that the effect seen in the Grail[−/−]Il21[−/−] and Grail[−/−] mice was due to the enhanced IL-21R signalling in the absence of Grail (Supplementary Fig. 10). Moreover, there were fewer IFN-γ[+] granzyme B[+] antigen-specific CD8[+] T cells from the tumours of Grail[−/−]Il21[−/−] mice compared to Grail[−/−] mice (Fig. 4h), supporting the role of IL-21R signalling in Grail[−/−] CD8[+] T-cell expansion and function. However, CD8[+] TIL from WT and Il21[−/−] mice had similar levels of functionality (Supplementary Fig. 10). Thus, the above data demonstrate that Grail deficiency results in increased IL-21R signalling in CD8[+] T cells and consequently enhanced cytolytic function against tumours in vivo.

**Grail regulates IL-21R expression**. Next we dissected the molecular mechanism for hyper-responsiveness of Grail[−/−] CD8[+] T cells to IL-21R signalling. IL-21 functions through IL-21R, which is a heterodimer that consists of γc (CD132) and IL-21Rα chain[17]. After binding of IL-21 to the IL-21Rα/γc complex, Janus-activated kinase 1 (JAK1) and JAK3 undergo autophosphorylation and subsequent phosphorylation of signal transducer and activator of transcription (STAT)-1 and STAT3 and to a lesser extent, STAT5. To determine which of the IL-21R signalling components is controlled by Grail; first, we checked the phosphorylation levels along with total protein levels of STAT1, STAT3 and STAT5 in naive WT and Grail[−/−] CD8[+] T cells after they were activated with anti-CD3 in the presence or absence of IL-21. TCR signalling alone did not induce phosphorylation of STAT1, STAT3 and STAT5 in either WT or Grail[−/−] CD8[+] T cells. Consistent with previous reports[18], IL-21 stimulation induced the phosphorylation of STAT1 and STAT3 in WT CD8[+] T cells (Fig. 5a). Compared to WT cells, Grail[−/−] CD8[+] T cells showed enhanced phosphorylation of these STATs as well as

STAT5 (Fig. 5a). However, the total levels of these STATs were comparable between WT and Grail[−/−] CD8[+] T cells (Fig. 5a), suggesting that molecules upstream of STATs account for the phenotype of Grail-deficient CD8[+] T cells. Next, we examined difference between IL-21-activated WT and Grail[−/−] CD8[+] T cells in JAK1 expression and activation, and observed augmented JAK1 phosphorylation in Grail[−/−] CD8[+] T cells upon IL-21 treatment, while total JAK1 level was comparable between WT and Grail[−/−] CD8[+] T cells (Fig. 5b).

Since STATs and JAK1 protein levels in CD8[+] T cells were not affected by Grail deficiency, we examined whether Grail controls the surface IL-21R expression. Interestingly, analysis of surface expression of γc/CD132 and IL-21Rα in naive CD8[+] T cells revealed increased protein levels of γC/CD132 and IL-21Rα level upon Grail deficiency (Fig. 5c). Since γc also constitutes the receptors for IL-2, IL-7 and IL-15, we checked expression of IL-7Rα, IL-2Rβ/IL-15Rb and IL-2Rα on naive CD8[+] T cells, and found that their expressions were comparable between WT and Grail[−/−] CD8[+] T cells (Supplementary Fig. 11a), suggesting that Grail specifically controls IL-21R levels on CD8[+] T cells. Consistent with receptor levels, Grail[−/−] CD8[+] T cells did not show enhanced proliferation and mRNA expression of IFN-γ and granzyme B in response to IL-2, IL-7 and IL-15 treatments (Supplementary Fig. 11b, c).

Next, we hypothesised Grail may regulate IL-21R protein level post-translationally through ubiquitination and degradation. To check whether Grail directly targets IL-21R subunits, we either overexpressed γc/CD132 or flag-tagged IL-21Rα together with WT or Grail mutant in 293T cells along with HA-tagged ubiquitin. After immunoprecipitation of γc/CD132 with CD132 antibody or IL-21Rα–Flag with Flag antibody and reprobing with HA-HRP antibodies, we observed the formation of high molecular weight ubiquitin conjugates in the presence of WT Grail, indicating polyubiquitin chain formation which was not observed on IL-15Ra (Fig. 5d; Supplementary Fig. 12). This ubiquitination was not present upon overexpression of the Grail mutant, which lacks a functional RING-finger domain (Fig. 5d, right lanes). Thus, IL-21R is a specific substrate for Grail-mediated ubiquitination in CD8[+] T cells. To determine whether Grail regulates the expression of IL-21R through endosome-mediated degradation, naive OT-I T cells from WT mice were activated with OVA peptide and WT APCs in the presence of the endosome inhibitor Latrunculin B. Addition of Latrunculin B led to upregulation of IL-21Rα and CD132 but not CD25, CD122 or IL-7R expression, suggesting that Grail may specifically target the endocytosed IL-21R complex via endosome-mediated degradation (Fig. 5e).

**Regulation of Grail expression by TGF-β**. We found that Grail expression levels were significantly upregulated in CD8[+] T cells infiltrated into the tumours compared to CD8[+] T cells that had homed to the spleens, indicating that tumour-specific factors

**Fig. 4** Grail[−/−] CD8[+] T cells exhibit a higher response to IL-21 than WT cells. **a** Real-time (RT)-PCR analysis of IL-21R expression in FACS-sorted and CD8[+] T cells from TILs after anti-CD3 restimulation. Results for target genes are presented after normalising to β-actin and shown as mean ± SEM. (n = 3). **b**, **c** Naive WT and Grail[−/−] CD8[+] T cells were activated with anti-CD3 alone or with IL-21. **b** Proliferation of WT and Grail[−/−] CD8[+] T cells was assayed 72 h after activation by adding [³H]-thymidine to the culture for the last 8 h. **c** Cells were harvested for RT-PCR analysis 24 h after activation. Results for target genes are presented after normalising to β-actin and shown as mean ± SEM. The mRNA expression level in WT CD8[+] T cells activated by anti-CD3 alone was set as 1. **d**, **e** Naive WT and Grail−/− CD8[+] T cells were stimulated with anti-CD3 in the presence or absence of IL-21 for 3 days. **d** IFN-γ and granzyme B (GzmB) protein production after determined by ELISA and flow cytometry. **e** Flow cytometry staining was performed for CD132 and IL-21Rα expression. **f**–**h** WT, Grail[−/−] and Grail[−/−]Il21[−/−] mice were inoculated with EG-7 tumour cells and analysed 17 days later. **f** Tumour size was measured and calculated as in Fig. 1a. **g**, **h** TILs were stained and analysed as described in Fig. 1. Results are shown on bar graphs as mean ± SEM as well as individual mice per group (n = 10 mice for WT and Grail[−/−]Il21[−/−] groups and n = 9 mice for Grail[−/−] group). All experiments were independently performed twice. *p < 0.05, **p < 0.01, ***p < 0.001 as determined using a Student's t-test. CPM count per minute

contribute to Grail expression in CD8$^+$ TILs. It has been reported that EL-4 cells suppress immune responses by secreting large amount of TGF-β[19]. Interestingly, we found that Grail is significantly upregulated in TGF-β-treated CD8$^+$ T cells (Fig. 6a). Thus, we further examined whether TGF-β signalling is responsible for Grail expression in tumour CD8$^+$ TILs by administering TGF-β blocking antibodies in WT mice during EG-7 tumour

growth. In fact, blockade of TGF-β resulted in a reduced rate of tumour growth and decreased tumour size at day 17 (Fig. 6b). Importantly, analysis of sorted CD8$^+$ TILs from both groups revealed a correlation between anti-TGF-β-mediated tumour blockade and diminished Grail (but not Cbl-b) expression in CD8$^+$ TILs (Fig. 6c). Furthermore, CD8$^+$ TIL from mice that received TGF-β blockade showed increased IL-21R expression by RT-PCR

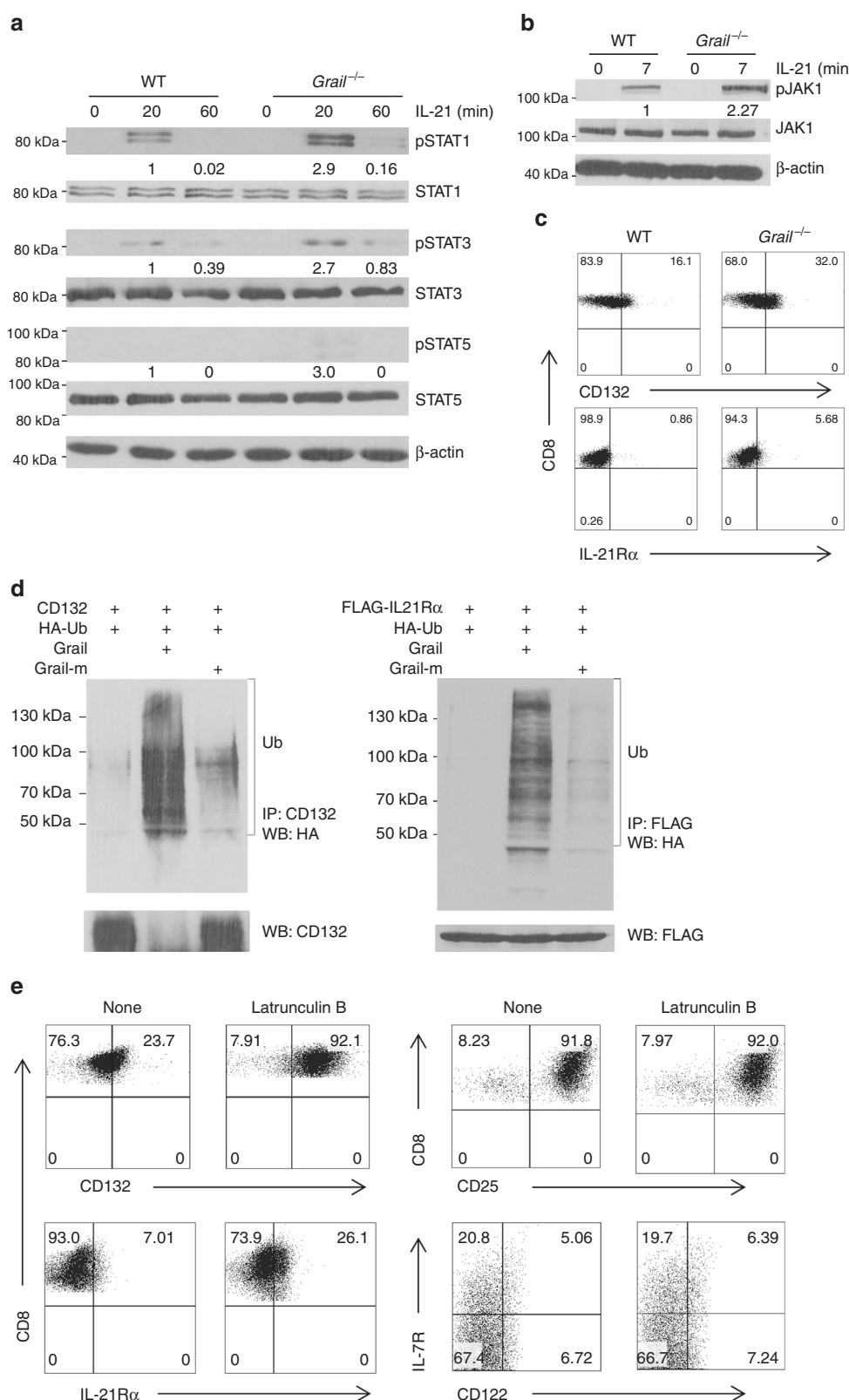

(Fig. 6d). Overall this data suggest that TGF-β signalling is one of the mechanisms responsible for Grail expression in CD8$^+$ T cells infiltrated into tumours and consequently to their unresponsiveness.

**Grail expression by CD8$^+$ T cells from patients with lymphoma**. The above data presented demonstrate a suppressive function of Grail in CD8$^+$ T cells in both an antigen-restricted and non-antigen-restricted mouse model of lymphoma. This led us to explore Grail and IL-21R expression patterns in CD8$^+$ T cells from the peripheral blood of patients with either diffuse large B-cell lymphoma (DLBCL) or follicular lymphoma (FL). To this end, peripheral blood mononuclear cells (PBMCs) were isolated from normal donors (ND PBMCs), DLBCL and FL, and stained for Grail expression by flow cytometry (Fig. 7a). Interestingly, Grail expression by CD8$^+$ T cells was significantly higher in patients with both subtypes of lymphoma as compared to ND PBMCs ($p = 0.024$, Fig. 7b). Similarly, to the murine models, the enhanced expression of Grail was specific to this E3 ubiquitin ligase as expression of Cbl-b was not different between ND and patient PBMCs (Fig. 7c). Furthermore, the level of IL-21R was significantly lost in the patient PBMC samples as compared to the ND (Fig. 7d). As levels of TGF-β present in the serum of patients with both FL and DLBCL have shown to be significantly higher than that found from ND, this could provide an explanation for the high level of Grail expression observed in the CD8$^+$ T cells present in the blood[20, 21]. Indeed, analysis of lymphoma patient serum samples showed significantly higher levels of TGF-β as compared to ND controls (Fig. 7e). Overall, this data suggest that the model described in the murine system where Grail expression is induced by TGF-β signalling and is involved in ubiquitination and degradation of the IL-21R may also occur in human patients with lymphoma. Further study of Grail in patients with lymphoma and solid tumours is warranted.

**Discussion**

To date, cancer immunotherapy is a promising approach to reactivate a patient's immune system to fight against tumours; however, it is a difficult process due to various tumour evasion mechanisms, particularly by T-cell tolerance induction. Thus, today it is of high priority to develop strategies to break T-cell tolerance and thereafter enhance T-cell cytotoxicity towards tumour cells. Previously, we have acknowledged the E3 ubiquitin ligase, Grail, as an essential component of CD4$^+$ T-cell tolerance[12, 13]. In the current study, we have determined the intrinsic role of Grail in controlling CD8$^+$ T-cell CTL functions. In fact, in CD8$^+$ T cells, Grail controls TCR and IL-21R signalling and restricts their effector cytokine expression and cytolytic function. As a result, Grail-deficient CD8$^+$ T cells are sufficient to control established tumours.

Our initial observation that Grail is preferentially upregulated in CD8$^+$ TIL compared to CD8$^+$ T cells localised in lymphoid tissues, suggested the potential role of Grail in CD8$^+$ T-cell anti-tumour activity. In fact, we observed a correlation between Grail expression in CD8$^+$ T cells and their unresponsiveness to tumours that was abolished upon Grail deletion. Importantly, we detected high levels of Grail expression in CD8$^+$ T cells from lymphoma patients compared to CD8$^+$ T cells from healthy donors, which further suggested that Grail could serve as a functional marker for nonresponsive CTLs. This hypothesis requires testing in a larger patient cohort and comparisons between liquid and solid tumours would be informative, especially as of TGF-β expression patterns could vary within the local tumour microenvironment. Surprisingly, we did not detect the upregulation of another E3 ubiquitin ligase, Cbl-b, in either CTLs infiltrated in mouse lymphoma or in CD8$^+$ T cells from lymphoma patients, indicating the distinct regulation and function of Grail and Cbl-b in CD8$^+$ T cells.

The ability of adoptively transferred Grail$^{-/-}$ CD8$^+$ T cells to control antigenic tumours provides evidence on the intrinsic role of Grail in controlling CD8$^+$ T-cell activation and function. Surprisingly, unlike Grail$^{-/-}$ CD4$^+$ T cells[12, 13], our in vitro and in vivo analysis showed that Grail deficiency did not change CD8$^+$ T proliferation and IL-2 production upon antigen encounter but did enhance resistance to Treg-mediated suppression. As Grail$^{-/-}$ Tregs also showed a weakened ability to suppress CD8$^+$ T cells, this deficiency could also play a role in the observed delay in tumour growth in the Grail$^{-/-}$ mice. However, production of the effector cytokine IFN-γ by Grail-deficient CD8$^+$ T cells and their cytolytic activity towards tumour cells were significantly enhanced, suggesting that Grail controls effector function of CD8$^+$ T cells. Since Grail selectively targets TCR-CD3 signalling for degradation in CD4$^+$ T cells[12], enhanced effector function of Grail$^{-/-}$ CD8$^+$ T cells could be attributed to sustained TCR signalling. We determined that Grail$^{-/-}$ CD8$^+$ T cells are defective in TCR downmodulation and furthermore Grail targets CD3ζ for ubiquitination and degradation, suggesting that sustained TCR-CD3 signalling is one of the mechanisms for excessive function of Grail$^{-/-}$ CTLs.

In addition to TCR engagement and positive costimulation, CD8$^+$ T cells require a third signal, cytokines for supporting their effector functions, survival and memory formation[22]. IL-21 is a cytokine produced by activated CD4$^+$ T cells and natural killer T cells[17] whose activity has been shown to stimulate CD8$^+$ T-cell cytotoxic function and survival[17, 23]. We found that Grail expression in patients with lymphoma CD8$^+$ T cells and mouse CD8$^+$ TILs negatively correlates with IL-21R expression while expression of other receptors involved in common gamma chain cytokine signalling (IL-2R, IL-7R and IL-15R) did not depend on the level of Grail. Our data also suggest the essential role of IL-21R signalling in Grail$^{-/-}$ CD8$^+$ T-cell function, since Grail$^{-/-}$ CD8$^+$ T cells in the absence of IL-21 signalling expressed significantly low level of cytolytic cytokines and failed to suppress tumour growth. Analysis of IL-21R downstream signalling components including Jak1, STAT1, STAT3 and STAT5 did not show any difference in their protein level upon Grail deficiency, while

---

**Fig. 5** Grail ubiquitinates the IL-21 receptor. **a** FACS-sorted naive WT and Grail$^{-/-}$ CD8$^+$ T cells were treated with or without IL-21 for the indicated time points (0–60 min). The whole cell lysate was subjected to western blot assay to detect the levels of phosphorylated and total STAT1, STAT3 and STAT5. β-actin was used as a loading control. **b** Naive WT and Grail$^{-/-}$ CD8$^+$ T cells were treated with or without IL-21 for 7 min, followed by western blot to detect the phosphorylated and total JAK1 levels. β-actin was used as a loading control. **c** γC (CD132) and IL-21Rα protein levels on the surface of naive untreated WT and Grail$^{-/-}$ CD8$^+$ T cells. Numbers in dot plot quadrants represent the percentages of each subset. **d** About 293T cells were transfected with vectors encoding CD132 or IL-21Rα, HA-Ub, and either Grail or a Grail mutant (Grail-m). The lysates were subjected to immunoprecipitation (i.p.) using an anti-FLAG or anti-CD132 antibodies. The blots were probed with anti-HA-HRP and re-probed with anti-FLAG or anti-CD132. **e** Naive WT and Grail$^{-/-}$ CD8$^+$ T cells were activated with OVA peptide and irradiated WT APCs alone or with Latrunculin B. γC (CD132), IL-21Rα, CD25, CD122 and IL-7R expression on CD8$^+$ T cells were assessed 48 h after activation. Numbers in dot plot quadrants represent the percentages of each subset. All experiments were independently performed three times

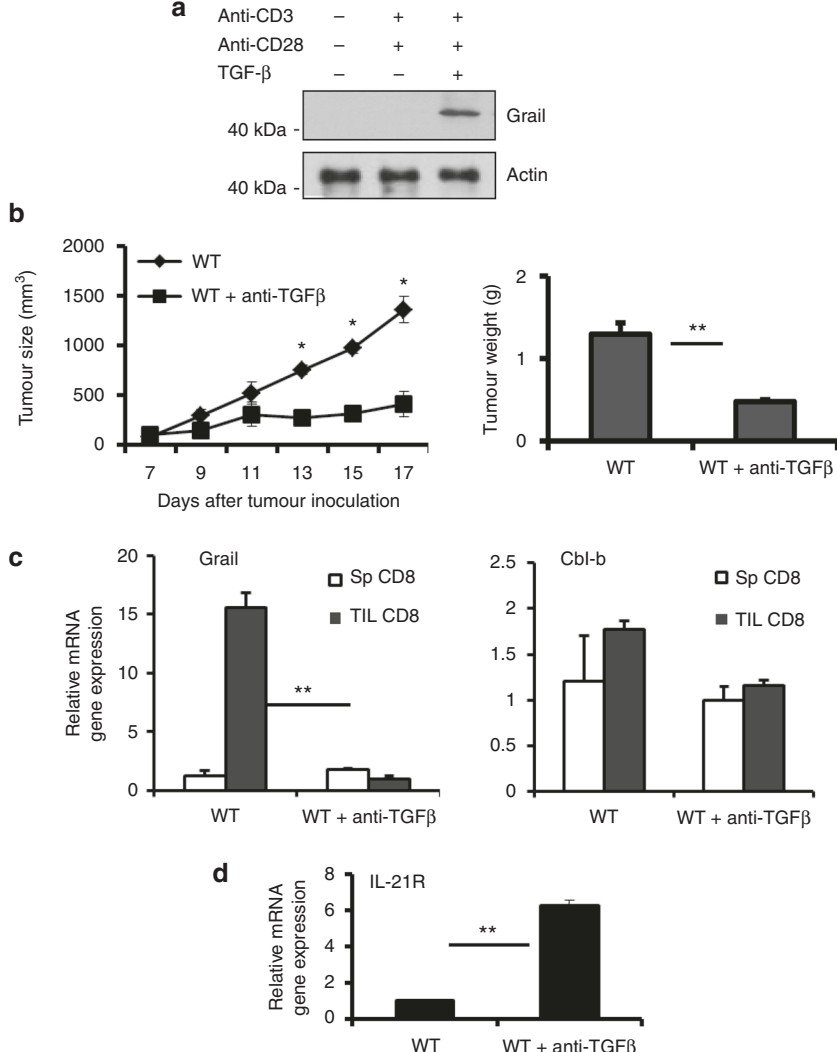

**Fig. 6** Regulation of Grail expression by TGF-β in CD8[+] TILs. **a** FACS-sorted naive CD8[+] T cells were activated with plate-bound anti-CD3/anti-CD28 in the presence or absence of TGF-β, followed by western blot to detect Grail levels. β-actin was used as a loading control. **b** WT mice were injected with rIgG or anti-TGF-β blocking antibodies on day 5, 7 and 9 after EG-7 inoculation and monitored daily for tumour growth. On day 17, tumours were isolated from each group and weight determined as in Fig. 1. ($n = 3$ independent experiments with five mice per group). **c**, **d** CD8[+] T cells from spleens and TILs were FACS-sorted and restimulated with plate-bound anti-CD3 for 4 h. The mRNA expression of Grail and Cbl-b (**c**) and IL-21R (**d**) were detected by RT-PCR analysis. Results for target genes are presented after normalising to β-actin and shown as mean ± SEM. The results shown are representative of three independent experiments. *$p < 0.05$, **$p < 0.01$ as determined using a Student's $t$-test

their phosphorylation levels were significantly enhanced. Interestingly, we found that protein levels of IL-21R subunits, γc/CD132 and IL-21Rα were already increased in naive Grail[−/−] CD8[+] T cells. Moreover, Grail specifically controls the expression of IL-21R subunits via ubiquitination and endosome-mediated degradation. Altogether, our study for the first time indicates that Grail controls the effector function of CD8[+] T cells by directly targeting and controlling IL-21R expression.

Overall, our results elucidate pivotal functions of Grail in the regulation of CD8[+] T-cell function. Sustained TCR-CD3 and IL-21R signalling helps Grail-deficient CD8[+] TILs to maintain their CTL activity. Thus modulation of Grail expression in CD8[+] T cells might be a potent new strategy to augment the effectiveness of tumour-specific CD8[+] T cells for anti-cancer immunotherapy.

## Methods

**Mice**. C57BL/6J, SJL, RAG1[−/−] and OT-I transgenic mice were purchased from Jackson Laboratories. Il21[−/−] mice, Grail[−/−] mice have been previously described[12, 24].

Grail[−/−] mice were crossed with OT-I mice or Il21[−/−] mice to get Grail[−/−] OT-I mice or Grail[−/−]Il21[−/−] mice, respectively. All mice are on a C57BL/6 background. About 8–10 week-old age- and sex-matched mice (either male or female) were used for experiments. Mice were housed in the SPF animal facility at M.D. Anderson Cancer Center and all animal experiments were approved by Institutional Animal Care and Use Committee.

**Cell lines**. EL-4 (murine lymphoma) and EG-7 were purchased from ATCC and maintained in DMEM medium supplemented with 10% fetal bovine serum and antibiotics. Cells lines were tested for mycoplasma contamination prior to any experiments.

**Patient samples**. All blood and tissue samples were obtained after written informed consent from patients through a University of Texas MD Anderson Cancer Center IRB-approved protocol (2005-0656).

**Flow cytometry**. To analyse the expression of Grail in human CD8[+] T cells, human PBMCs were first stained for surface antigens (CD3, CD8) and a live/dead dye followed by fixation and permeabilized for staining with Grail. The following antibodies were obtained from BD Biosciences: PerCP-Cy5.5-conjugated anti-CD3 (clone SK7, 1:20 dilution), Pacific blue-conjugated anti-CD8 (clone RPA-T8, 1:50

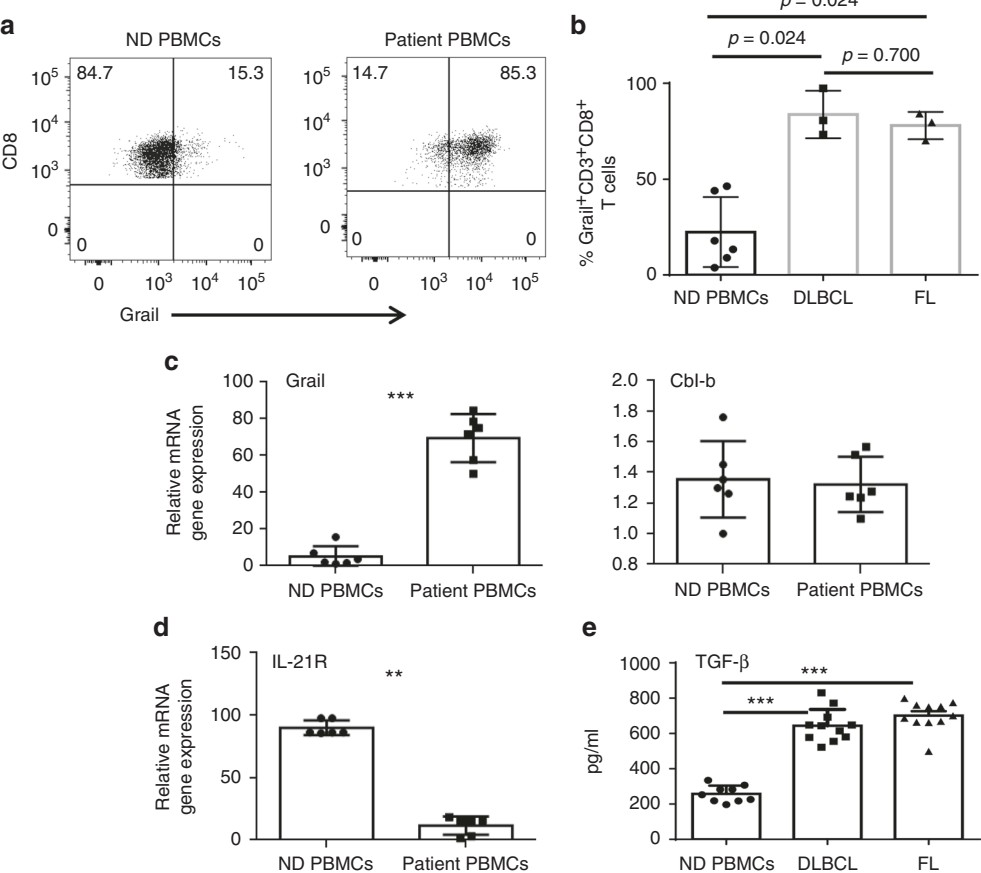

**Fig. 7** CD8$^+$ T cells from patients with lymphoma express higher levels of Grail compared to normal donors. **a–d** PBMCs collected from normal donors (ND) and lymphoma patients ($n = 6$ for both groups) were stained with antibodies against CD3, CD8 and Grail. Dead cells were discriminated using AQUA live/dead dye. Gating for Grail was performed using fluorescence minus one and secondary antibody alone controls. **a** Grail expression from ND PBMCs (*left*) and patient PBMCs (*right*) is shown. Cells are gated on live, CD3$^+$CD8$^+$ T cells. **b** Grail expression in CD3$^+$CD8$^+$ T cells from normal donors ($n = 6$) and lymphoma patients ($n = 3$ DLBCL and $n = 3$ FL). **c, d** CD3$^+$CD8$^+$ T cells were FACS-sorted and mRNA expression of Grail, Cbl-b (**c**) and IL-21R (**d**) was detected by RT-PCR analysis. Results for target genes are presented after normalising to β-actin. **e** Serum from lymphoma patients and normal donors was assessed for levels of TGF-β by ELISA. ($n = 9$ normal donors, 11 DLBCL and 11 FL patients). **b, e** *Bar graphs* shows mean ± SEM as well as individual patient samples. **$p < 0.01$, ***$p < 0.001$ as determined using a Student's t-test

dilution). Alexa488-conjugated goat anti-rabbit IgG (Jackson ImmunoResearch, 1:300 dilution). Rabbit monoclonal Grail antibody (anti-RNF 128, 1:100 dilution) was obtained from Abcam. Live/dead Aqua dye (1:200 dilution) was purchased from Invitrogen. For animal experiments, the following antibodies were purchased from Biolegend: CD8 PerCP-Cy5.5 (clone 53-6.7, 1:1000 dilution), CD4 FITC (clone GK1.5, 1:100 dilution), CD25 PE (clone PC61, 1:100 dilution), CD62L FITC (clone MEL-14, 1:800 dilution) CD44 APC (clone IM7, 1:500 dilution), CD69 PE (clone H1.2F3, 1:100 dilution), CD132 PE (clone TUGM2, 1:100 dilution), CD127 PE (IL-7R, clone CB1199, 1:100 dilution), Rag IgG2ak biotin (clone RTK2758, 1:100 dilution), Annexin V FITC (1:20 dilution), TCRβ FITC (clone H57-597, 1:100 dilution), OX-40 APC (clone OX86, 1:100 dilution) and Bcl-2 Alexa Fluor 647 (clone Bcl/10C4, 1:100 dilution). Antibodies were also purchased from BD Biosciences: granzyme B Alexa Fluor 647 (clone GB-11, 1:100 dilution), IL-2 APC (clone JES6-5H4, 1:100 dilution), 7AAD (1:20 dilution) and caspase-3 PE (C92-605, 1:5 dilution). Antibodies from eBiosciences were also purchased: IL-21R biotin (clone eBio4Ag, 1:100 dilution), IFNγ PE (clone XMG1.2, 1:200 dilution), CD122 PE (clone IM-b1, 1:100 dilution) and 4-1BB PE (clone 17B5, 1:100 dilution). Streptavidin APC (1:500 dilution) was purchased from Jackson ImmunoResearch. Gating was determined using the Fluorescence minus one approach. Samples were acquired using a BD FACs Canto and analysed using FlowJo v 10.0.7. The gating strategy for CD8$^+$ T-cell phenotyping is described in Supplementary Fig. 13.

**Detection of serum TGF-β.** TGF-β protein levels in serum of healthy individuals and lymphoma patients were detected by enzyme-linked immunosorbent assay (ELISA) using the manufacturer's instructions (eBioscience).

**T-cell function analysis in vitro.** Naive WT or Grail$^{-/-}$ CD8$^+$ T cells (CD8$^+$CD25$^-$CD62L$^{hi}$CD44$^{lo}$) were flow cytometry-sorted and activated with plate-bound indicated doses of anti-CD3 and/or anti-CD28 (eBioscience). Naive OT-I or Grail$^{-/-}$

OT-I cells (CD8$^+$CD25$^-$CD62L$^{hi}$CD44$^{lo}$) were sorted and activated with indicated doses of OT-I peptide (SIINFEKL) (RS Synthesis) in the presence of irradiated WT APCs. IL-2 production was determined 24 h after T-cell activation by ELISA using manufacturer's instructions (BD Biosciences). Proliferation was assayed after 2 days (for 0.001 µg/ml OVA stimulation) or 3 days (for anti-CD3 (2 µg/ml, clone 2C11, eBioscience) and/or anti-CD28 (2 µg/ml, clone 37.51, eBioscience) stimulation) of activation by adding [3H] thymidine (Perkin Elmer) to the culture for the last 8 h. IFN-γ and granzyme B production was analysed by flow cytometry 2 days (OVA stimulation) or 3 days (anti-CD3 and/or anti-CD28 stimulation) later. In Fig. 4b, c, FACS-sorted naive CD8$^+$ T cells from WT and Grail$^{-/-}$ mice were activated with plate-bound anti-CD3 (2 µg/ml, eBioscience) alone or together with IL-21 (50 ng/ml, Peprotech). Cells were harvested 24 h later for gene expression analysis. About 72 h after activation, proliferation was assayed by adding [3H] thymidine (Perkin Elmer) and IFNγ and granzyme B were detected by ELISA using manufacturer's instructions (BD Bioscience) and flow cytometry staining. For Treg suppression assay, naive CD8$^+$ T cells were cultured with plate-bound anti-CD3 and irradiated WT APCs or together with WT or Grail$^{-/-}$ Foxp3-GFP$^+$CD4$^+$ nTreg cells in triplicate wells. Proliferation was analysed 72 h after treatment by adding [3H] thymidine (Perkin Elmer) to the culture for the last 8 h. In Supplementary Fig. 10c, FACS-sorted naive CD8$^+$ T cells from WT and Grail$^{-/-}$ mice were activated with plate-bound anti-CD3 (2 µg/ml, eBioscience) alone or together with hIL-2 (30 U/ml, Peprotech) or mIL-7 (5 ng/ml, Peprotech) or mIL-15 (5 ng/ml, Peprotech) and 24 h later cells were harvested for gene expression analysis.

The in vitro SIINFEKL-specific CTL response was measured using a caspase-3 cleavage assay[25]. In brief, naive CD8$^+$ T cells from WT and Grail$^{-/-}$ OT-I mice were activated with OT-I peptide (0.001 µg/ml; RS Synthesis) and irradiated WT APCs for 2 days. EL-4 cells were used as target cells after labelling with DDAO-SE and 1 µg/ml peptide. Activated CD8$^+$ T cells were co-cultured with the target EL-4 cells at a 1:1 ratio for 2 h at 37 °C. The cells were permeabilized and stained with PE-conjugated anti-cleaved caspase-3 antibody. The percentage of cleaved caspase-

3-positive cells among DDAO-SE-labelled target cells were analysed by flow cytometry.

The in vivo SIINFEKL-specific cytolytic activity of CD8[+] T-cell responses was measured as follows. Briefly, spleen cells from OT-I or Grail[−/−] OT-I mice were adoptively transferred (iv) into C57BL/6J mice (6–8 week old, 10[6] cells per mouse) followed by s.c. vaccination with OVA peptide (100 μg per mouse, RS Synthesis) and anti-CD40 (50 μg per mouse, clone FGK4.5, BioXcell). Imiquimod cream 5% (50 μg/mouse, Aldara, Fougera) was applied on the vaccination site. In addition, mice intraperitoneally (i.p.) received 100,000 IU rhIL-2 (TECIN, Hoffman LaRoche Inc.) as previously described[26]. Three days after vaccination, mice were injected with a 1:1 mix of target cells (5 × 10[6] of each). Target cells were prepared using splenocytes from C57BL/6 mice either loaded with 1 μg/ml peptide and labelled with 5 μmol/l CFSE or unloaded and labelled with 0.5 μmol/l CFSE. Four and 8 h later, splenocytes from the recipients were analysed by flow cytometry to assess peptide-specific killing of the CFSE[hi] labelled cells.

**Quantitative real-time PCR.** Total RNA was prepared using TriZol reagent (Life Technology). Complimentary DNA (cDNA) was made using Superscript reverse transcriptase and oligo(dT) primers (Life Technology), and gene expression was detected with a Bio-Rad iCycler Optical System using iQ SYBR green RT PCR kit (Bio-Rad Laboratories, Inc.). The data were normalised to the reference gene β-actin. The primers for Grail, IFN-γ, TNF, IL-21, IL-21R, granzyme B, Cbl-b, OX-40, 4-1BB, Bcl-xL and β-actin were previously described[12, 16, 24, 27–36]. Briefly, the pairs used for mouse are: Grail forward: 5′-TAGCTGTGCTGTGTGCATTG-3′, reverse: 5′-CTTCATGGGGAGAGGCAGTA-3′; IFN-γ forward: 5′-GATGCATTC ATGAGTATTGCCAAGT-3′, reverse: 5′-GTGGACCACTCGGATGAGCTC-3′; TNF-α forward: 5′-AATGGCCTCCCTCTCATCAGT-3′, reverse: 5′-GCTACAG GCTTGTCACTCGAATT-3′; IL-21 forward: 5′-TCCCACGCCCAGGAGAC CACC-3′, reverse: 5′-ATGCTGGGGTTGGGGCTCCGT-3′; granzyme B forward: 5′-ATCAAGGATCAGCAGCCTGA-3′, reverse: 5′-TGATGTCATTGGAGAA TGTCT-3′; Cbl-b forward: 5′-TTCCAGATGGCAAACTCAATG-3′, reverse: 5′-TACATTCTCTCTCCTTGCCTTCTTTA-3′; OX-40 forward: 5′-AACCTCG GCAGGACAGCGGC-3′, reverse: 5′-CACTGGCTGGGTGGCGGGTC-3′; 4-1BB forward: 5′-CCCCCACATATTCAAGCAAC-3′, reverse: 5′-TAGCCTCCTC CTCCTCCTTC-3′; Bcl-xL forward: 5′-TGGTGGTCGACTTTCTCTCC-3′, reverse: 5′-CTCCATCCCGAAAGAGTTCA-3′; perforin 1 forward: 5′-AGG-CAAACATGCGCGCCTCCGT-3′, reverse: 5′-GGACCGAGATGCGGCCACC GA-3′; b-actin forward: 5′-TGGAATCCTGTGGCATCCATGAAAC-3′, reverse: 5′-TAAAACGCAGCTCAGTAACAGTCCG-3′. For human samples the following pairs were utilised: Grail forward: 5′-CTGCTCGAAGGCTACGGAAT-3′, reverse: 5′-GGGCCAATTTCCTTGTCTCCT-3′; IL-21R forward: 5′-GGGCTCTGTGAT GTAGGCAG-3′, reverse: 5′-CTGGTCTTGCCAGGTAAGGG-3′; Cbl-b forward: 5′-TGTCTCTGGACAGCTACGGC-3′, reverse: 5′-GTCTTACCACTTTGTCCAT GAGC-3′; actin forward: 5′-TCCCTGGAGAAGAGCTACGA-3′, reverse: 5′-AGC ACTGTGTTGGCGTACAG-3′.

**Immunoblot analysis and ubiquitination assay.** Cells were lysed in kinase assay lysis buffer supplemented with protease inhibitor cocktail (Roche) and phosphatase inhibitors (10 mM NaF and 1 mM Na3VO4). Protein concentration was determined by Bio-Rad Bradford protein assay and equal amounts of protein were loaded for immunoblot analysis with antibodies against phospho-STAT1 (Tyr701), phospho-STAT3 (Ser727), phospho-STAT5 (Tyr694), phosphor-JAK1 (Tyr1022/1023), STAT1, STAT3, JAK1 (Cell Signaling), STAT5 (Santa Cruz Biotechnology), Grail (Abcam) (1:100 dilution). Anti-β-actin (1:1000 dilution), anti-mouse IgG-HRP (1:5000 dilution) and anti-rabbit IgG-HRP (1:5000 dilution) secondary antibodies were from Thermo scientific. β-actin was used as the loading control for all experiments.

For the ubiquitination assays, 293 T cells were transfected in six-well plates with bicistronic retroviral expression vector pGFP-RV containing either IL-21Rα, CD132 or IL-15Rα, and Grail or Grail mutant and pcDNA-HA-ubiquitin. Cells were lysed in kinase lysis buffer supplemented with N-ethylmaleimide (NEM). Lysates were incubated with anti-FLAG M2 mAb (Sigma Aldrich), anti-CD132 (Santa Cruz) or anti-IL-15Rα (Santa Cruz) (5 μg per sample) and then Protein A-agarose (Sigma Aldrich), followed by immunoblotting using anti-HA-HRP, anti-IL-15Ra, anti-CD132 (Santa Cruz Biotechnology) and anti-Flag antibodies (Sigma) (1:1000 dilution). Uncropped gel images are shown in Supplementary Fig. 14.

**EL-4 and EG-7 tumour models.** 1 × 10[6] EG-7 or EL-4 cells were inoculated s.c. to the right flanks of experimental animals (8–10 week old) on day 0 and tumours were measured every other day using a digital caliper from day 7. Tumour sizes were calculated by ½(ab[2]), where a is the length and b is the width. The maximum tumour size allowed and adhered to is 1.5 cm tumour diameter. For the adoptive transfer therapeutic model, congenic SJL (8–10 week old) recipient mice were inoculated with EG-7 cells and 5 days later when tumours were palpable transferred with 3 × 10[6] OT-I or Grail[−/−] OT-I cells by i.v. In some experiments, purified OT-I and Grail[−/−] OT-I cells were labelled with 5-(and 6)-CFDA SE, CFSE before being transferred. For CD8[+] T-cell depletion experiments, 150 μg anti-CD8 (clone 2.43, BioXcell) was given by i.p. 3 days before tumour inoculation and weekly after that until the end of experiments. Rat IgG (BioXcell) was administered to control mice. The depletion efficiency was checked by blood staining with corresponding

antibodies for each experiment. For TGF-β blockade, 100 μg anti-TGF-β (clone 1D11, BioXcell) was given 5 days after EG-7 inoculation three times with 2 day intervals. Rat IgG (BioXcell) was administered to control mice.

**Statistical analysis.** Graphical presentation and statistical analysis of the data were performed using GraphPad Prism (Version 5.0d, GraphPad software, San Diego, CA, USA) and Excel. Data are displayed as mean and SEM. Results between experimental groups were compared using a one-way analysis of variance or Student's t-test. $p < 0.05$ was considered statistically significant. Statistical significance is displayed as $*p < 0.05$, $**p < 0.01$, $***p < 0.001$ and $****p < 0.0001$.

**Data availability.** The authors declare that the data supporting the findings of this study are available within the article and its supplementary information files, or are available from the corresponding author upon request.

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

## Acknowledgements

We thank the members of Dr. Chen Dong laboratory, as well as Dr. S.-C.S. laboratory for discussions; and the FACS core facility at MD Anderson Cancer Center for assistance. This work is supported by NIH research (AI083761 (R.N.), A1R03AI120027 (R.N.), 1R21AI20012 (R.N.), AI064639 (S.-C.S.), CA184845 (C.B.), CA178580 (C.B.), CA155143 (S.S.N.) and P30CA16672 (flow core)), Institutional Research Grant (R.N.), start-up grant (R.N.), MD Anderson CIC seed grant (R.N.). L.W.K. is recipient of Leukemia & Lymphoma Society Quest for Cures grant (0855-14). K.R. is recipient of CPRIT Fellowship (RP101503). The primary blood and tumour samples were provided by The University of Texas MD Anderson Cancer Center Lymphoma Tissue Bank, which is supported by the Lymphoma SPORE P50CA136411 (NIH), Cancer Center Support Grant CA16672 (NIH) and Fredrick B Hagemeister Research Fund.

## Author contributions

R.N.: Formulated the hypothesis, designed all experiments and wrote the manuscript. Y.Y. and C.H.: Planned and performed most of the experiments. J.W.: Initiated the study. Q.Z. and D.S.: Assisted with western blotting experiments. O.N.H.: Helped with flow cytometry staining. Y.H.: Helped with *in vivo* killing assay. S.S.N.: Provided serum, blood and tissue samples from lymphoma patients. K.R.: Performed analysis of human samples. K.S., T.W., W.W.O., C.B., H.Q., L.W.K. and S.-C.S.: Helped with discussion and critical reading of the manuscript. A.S. and A.A.: Assisted with manuscript writing and cloning, respectively.

## Additional information

**Competing interests:** The authors declare no competing financial interests.

