## [Peer Review File · Nature Communications]

Reviewers' comments:

Reviewer #1 (Remarks to the Author):

In this manuscript, the authors investigate the role of the Grail protein in the function of CD8 T cells using a tumor model. Previously this group has characterized the function in CD4 T cells, Th2 cells and T reg cells using the Grail KO mouse. It is suggested that Grail deficiency leads to more efficient tumor killing and that it might be a good target for upregulating CTL activity. Experiments attempt to connect Grail expression to IL-21R expression fail to define the functional outcome of this relationship. In addition, the *in vivo* functional cytolytic activity of Grail deficient CD8 T cells is not assessed.

Specific comments:

Fig. 1: Do Grail KO CD8 T cells accumulate because the tumor is dying? A time course of TIL accumulation in the WT and Grail KO mice would begin to address this.

Although *in vitro* stimulation of TILs reveals large differences in Gzmb/Ifng expressors, when CD8 TILs are sorted from tumors, the differences in mRNA expression of cytolytic factors are actually quite small.

Experiments presented here beg the question of whether Grail KO CD8 T cells are actually better killers.

Fig. 2: The CD8 transfer experiments demonstrate much better tumor elimination *in vivo* with the Grail KO cells, but the CD4 and CD8 accumulation in the tumor are not greatly increased. In Fig 2e, there are actually more Gzmb+ cells in the WT than in the Grail KO. Again, are the Grail KO cells really better killers *in vivo* or is this an indirect effect?

Fig. 3: In the bone marrow chimera experiments, again there are more Gzmb+ cells in the WT than in the Grail KO. The effect on Gzmb+Ifng+ CD8 is very small.

Fig 3c-e show effects on naïve sorted CD8 T cells *in vitro*. The induction of Gzmb+Ifng+ cells is quite substantial and the cytolytic activity against OT1 pulsed EL-4 is higher, but this *in vitro* assay does not necessarily translate to an *in vivo* killing effect in the tumor.

Fig. 4: It is not clear why the role of IL-21 in Grail inhibition of CD8 killing was investigated. Fig 4a-c document *in vitro* effects of IL21 on WT and Grail KO CD8 T cells, but this is not sufficient to connect the Grail effect to the IL-21 effect. In the tumor experiment shown in Fig. 4d, although lack of IL-21 does away with the enhanced tumor killing in the Grail KO, the tumor curve for the IL-21 KO is not included-is it perhaps worse than the WT, in which case the IL-21 effect and the Grail effect are distinct.

Fig. 5: Original flow data should be displayed (5c)-it is unlikely that naïve CD8 T cells express high levels of IL-21R. The ubiquitination of IL-21R components in 293T cells should not be interpreted as proof of a role of ubiquitination in CD8 T cells.

Fig. 6: Although TGFb clearly upregulated Grail protein in activated CD8 T cells *in vitro*, blocking TGFb *in vivo* was not shown to affect Grail protein expression in TILs, nor was it shown to impact tumor growth.

Fig. 7: It is enticing to believe that enhanced Grail expression in the patient CD8+ T cells is a result of higher TGFb levels and that it leads to reduced IL-21R levels, but no causal relationship is demonstrated here.

Reviewer #2 (Remarks to the Author):

This work is based on the knowledge that the E3 ubiquitin ligases Itch, Cbl-b and Grail may be involved in the regulation of T cell function. The authors examined Grail in murine and human CD8 T cells. They used EL4 and EG-7 cells as transplantable tumor model and also briefly studied CD8 T cells from lymphoma patients. Higher levels of Grail were found to be associated with lower levels

of IL-21R in both species. The mouse experiments further show a role for TGF β in the upregulation of Grail and reduced IL-21R expression. In absence of Grail, CD8 T cells produced more IFN γ and GrzmB and conferred better antitumor effects in vivo.

The study is based on a meaningful hypothesis. However, the data is somewhat unclear, only based on a transplantable tumor model, and very limited results from patient's PBMC.

Increased GzmB production in absence of Grail does not seem to be very consistent, when focusing on Figure 2e that shows that Grail $^{-/-}$ OT-1 cells produced more IFN γ but not more GrzmB than OT-1 cells. The same applies for Figure 3c, right side. These results are different as compared to the data shown in Figure 1d and other parts of Figure 3. Nevertheless, the authors state that the data shown are representative.

In the dot plots of Figure 3a, the gating and the percentages of GzmB \pm cells is not shown, which is inconsistent to the remaining data on this subject. On several occasions, it is difficult to explain what the authors mean with "representative data" when comparing the dot plots with the histograms. I suggest that the authors provide all the values of all experiments, which can also be done e.g. as tables and/or as supplementary material.

Supplementary Figure 5 only shows a single dot plot, without x- and y-axis labels, and there is no a) and b) in this Figure. Thus, it is not possible to review the data on TCR expression.

In the first part, the authors state that Grail has no effects on CD8 T cell proliferation. However, subsequently (Figure 4 and text), the data show different proliferation and the authors conclude on "expansion".

The authors repetitively state that "Grail-deficient CD8 $^{+}$ T cells are sufficient to (completely) reject established tumors". However, the majority of data show that tumors were not completely rejected.

Also at several other occasions, "greatly increased" or similar wording does not reflect the often relatively small differences, suggesting over-interpretation of the findings. Furthermore, the multiple roles of Grail, beyond the ones studied here, are insufficiently investigated.

The data provided from lymphoma patients is promising but very limited. Additional experiments assessing T cell function are necessary in order to justify that the author's conclusion also apply to humans.

Minor point: "TILs" may be replaced with "T cell" in the first subtitle of the results section ("Grail loss correlates with enhanced CD8 $^{+}$ TILs infiltration in lymphoma tumors")

Reviewer #3 (Remarks to the Author):

The manuscript by Wang et al investigates the role of Grail in anti-tumour immunity mediated by CD8 $^{+}$ T cells. While their observations are potentially interesting, the quality of the work is a concern. The major issues of concern are listed below:

The majority of the experiments were only performed twice and, almost invariably, the authors show only representative data. In some instances, it appears as though the experiments were only done once, e.g. Fig. 5A, B, D, while the number of replicates in experiment C was unclear. In another example, the authors state in the legend for Figure 4 that "the expression level in WT cells was set as 1". Where does the error bar in WT samples come from then? There are many other uncertainties of a similar kind found in most of the figures.

The experimental protocols are inconsistent. For example, in Figure 3, the T cells were activated for different periods of time: 3 days (3C), 2 days (3D) and 7 days (3E). In a similar experiment shown in Suppl Fig. 6B, the cells were activated for 24hrs. Given the disparity in the activation periods between experiments, the interpretation of these results must be questioned.

I also question the design of the in vivo experiments, where all the mice were culled on Day 17 - the day that WT mice had to be culled due to high tumour burden. At that point, Grail $^{-/-}$ animals could still control (but not reject) the tumours. These mice should be observed for at least another 5-7 weeks to demonstrate the prolonged/constitutive anti-tumour activity of Grail $^{-/-}$ T cells (or

tumour rejection). As shown by the authors, tumours can "overwhelm" the animals within less than a week. Therefore, the data presented does not provide sufficient information to justify the conclusion of this part of the study.

TILs were only analyzed on Day 17, when WT mice had to be culled due to uncontrolled tumour growth. Had the TILs been assessed earlier, when the tumours in WT and Grail^{-/-} animals were still small and remained under the putative control of the immune system - e.g. on Days 9-11 - it would perhaps be possible to determine a mechanism that allowed Grail^{-/-} mice to control tumour growth. For the authors to interpret this data adequately, and draw solid conclusions, the TILs would need to be analyzed at various time points during the course of tumour growth, particularly during the critical early period of immune control.

The experimental procedures are also poorly described. For example, anti-CD3/CD28 or SIINFEKL stimulation of the T cells had to be done in the presence of IL-2 (this was not stated in the M&M). If so, how can the results in Suppl Fig 6B (how many times was this experiment done?) showing the identical IFN-gamma mRNA levels following IL-2 treatment (concentration is unknown) be reconciled with a large difference in the intracellular levels of IFN-gamma between WT and Grail^{-/-} shown in Figure 3C,D. I understand that these are two different readouts (although done on different days), but if so, what is the biological relevance of mRNA levels shown in Figures 1, 4, 6, 7? This discrepancy should be addressed in the manuscript.

Reviewer #4 (Remarks to the Author):

Whereas the role that Grail has as a regulator of CD4 T cell activation has been well characterized, little was known about whether or how this E3 ubiquitin ligase might control CD8 T cell function. In this study using a model of s.c. transfer of lymphoma, the authors identify Grail as a regulator of the anti-tumor CTL response. The data show that CD8 TILs upregulate the expression of Grail where it likely dampens effector functions and responses to IL-21, as CD8 T cells deficient in Grail show increased IFN γ expression and increase response to IL-21 that results in enhanced control of tumor growth. The paper identifies, thus, a novel and very relevant regulator of anti-tumor CD8 T cell responses and should help us understand better the mechanisms that mediate tumor-induced immune evasion.

There are a series of points that would need to be addressed, though, to fully support some of the conclusions reached by the authors and better understand how Grail controls CTL function. Based on the data presented, the conclusion reached by the authors is that Grail is not involved in the direct regulation of CD8 T cell activation but specifically in the regulation of effector cytokine expression by those cells. The data supports for the most part this assertion but there are a few points that need clarification. While all the FACS data is described as changes in the number of GrzB+IFN γ + CD8+ T cells, many FACS analyses show little if any difference in the expression of GrzB (e.g. Figs 2c-e; Fig 3 b-d...) and the increased in the % of double positive cells in those experiments reflects merely and increased in the cells expressing of IFN γ . Would IFN γ be the more specific target of Grail regulation rather than both IFN γ and GrzB? Aside for the studies on IL-21R signaling (see comments below) one possible proposed mechanism that may explain the role of Grail is its ability to modulate TCR downregulation. If this is true, how would the authors explain those changes in availability of TCR can affect the expression of some cytokines (i.e. IFN γ) and not others (i.e. IL-2)?

The other proposed target of Grail is the IL-21R. The authors show increased sensitivity of CD8+ T cells that lack Grail to IL-21 and loss of the ability of Grail^{-/-} mice to control EG7 growth in IL-21 deficient mice. The authors proposed that Grail-mediated ubiquitination of the common gamma chain and the IL-21R α chain would regulate their degradation, which would account for the increased expression of the IL-21R in Grail deficient CTLs. However, the paper does not show any experiment to provide evidence of this increased turnover. This is further complicated by the fact that there is a decreased expression of the mRNAs for both genes. Does Grail also control gene expression of the IL21R subunits or are those changes a result of some positive feedback? The data on ubiquitination in the 293 system is also a little difficult to interpret. Multiple bands that do not appear to correlate with multiples of Ub appear on the IPs which may suggest some problems with the IP in an overexpression system. It would help if it could be shown for instance that in this system, one a specific chain of one of the other common gamma chain cytokine receptor that do not appear to be affected by Grail, is indeed not ubiquitinated when Grail is overexpressed.

The experiments that identify TGF β as a possible regulator of Grail expression in CTLs are very interesting as they could help explain how the expression of this protein is upregulated in a tumor

microenvironment frequently rich in TGF β . It would be interesting, though, to verify that the increased in Grail caused by TGF β is enough to downmodulate the expression of the IL-21R in those cells.

Other points:

Representative FACS images for Fig. 5c should be shown.

In several instances it is claimed that transfer of Grail $^{-/-}$ CD8 $^{+}$ T cells is sufficient to mediate rejection of established EG-7 tumors to support a possible therapeutic value. However, in those experiments cells were transfer to animals on day 5 when tumors are still not detectable. It might be a good idea to more precisely define those experiment or alternatively analyze the possible effect of CTL transfer when tumors are already detectable (day 9-10?)

In Fig 3b, it would be interesting to complete the data by showing any differences in the % of CD45.1 and CD45.2 CTLs that infiltrate the tumors.

Regarding the differences in expression of mRNAs for cytokines in sorted TILs shown in Fig. 1e, though they appear statistically significant, it is difficult to clearly asses a biological meaning to increases in gene expression that seem to range from 30 to 60% (all less than 2-fold). This is likely the result of the limited % of T cells that might be expressing those cytokine genes at a given time in vivo, but I am not sure the data as it is can support biological relevance of those increases.

Responses to reviewers' comments

We thank all reviewers for their critiques, comments and helpful advice. We have modified the manuscript according to the recommendations. Changes are indicated by underlines in the revised manuscript. Below are the answers addressing the critiques raised by the reviewers.

Reviewer #1:

In this manuscript, the authors investigate the role of the Grail protein in the function of CD8 T cells using a tumor model. Previously this group has characterized the function in CD4 T cells, Th2 cells and T reg cells using the Grail KO mouse. It is suggested that Grail deficiency leads to more efficient tumor killing and that it might be a good target for upregulating CTL activity. Experiments attempt to connect Grail expression to IL-21R expression fail to define the functional outcome of this relationship. In addition, the in vivo functional cytolytic activity of Grail deficient CD8 T cells is not assessed.

-We would like to thank the reviewer for the comments and critiques. We have attempted to make a better connection between Grail expression and IL-21R expression and signaling by clarifying the text and providing additional experimental evidence in **Figure 5e**. In addition, we have designed and carried out an *in vivo* killing assay to demonstrate the enhanced CTL function of Grail deficient CD8⁺ T cells in **Figure 3f**.

Specific comments:

Fig. 1: Do Grail KO CD8 T cells accumulate because the tumor is dying? A time course of TIL accumulation in the WT and Grail KO mice would begin to address this.

-We agree that this is an excellent point and to this end, we performed a time course analysis of TIL accumulation in the tumors of both WT and *Grail*^{-/-} mice. We did not detect differences in the percentage of tumor-infiltrated CD4⁺ and CD8⁺ T cells between WT and *Grail*^{-/-} groups on day 7 after tumor inoculation; however, by day 12 we observed that the percentage of CD8⁺ TIL was significantly increased in *Grail*^{-/-} mice as compared to WT mice. To explore the possible mechanism(s) for accumulation of *Grail*^{-/-} CD8⁺ T cells in the lymphoma, first we analyzed the susceptibility of WT and *Grail*^{-/-} CD8⁺ TILs to apoptosis; however, we did not detect any difference in the percentage of apoptotic markers Annexin V and 7-AAD in CD8⁺ TIL between WT and *Grail*^{-/-} groups on days 7 and 12 after tumor inoculation. In addition, we analyzed the expression of pro-survival costimulatory receptors OX-40 and 4-1BB and anti-apoptotic factor Bcl-2 in CD8⁺ WT and *Grail*^{-/-} TIL on days 7 and 12 after tumor inoculation. Interestingly, we found increased expression of OX-40, 4-1BB and Bcl-2 by *Grail*^{-/-} CD8⁺ TIL on days 7 and 12. Moreover, mRNA levels of pro-survival costimulatory molecules OX-40 and 4-1BB and anti-apoptotic gene Bcl-xL (B-cell lymphoma-extra large) were enhanced on day 17 as well, suggesting that Grail could control the survival of CTLs infiltrated into the tumors and thus result in the accumulation of this subset within the tumor. The additional data has been added to **Figure 1b** and **Supplementary Figure 4a**.

*Although in vitro stimulation of TILs reveals large differences in *Gzmb*/*Ifng* expressors, when CD8 TILs are sorted from tumors, the differences in mRNA expression of cytolytic factors are actually quite small. Experiments presented here beg the question of whether Grail KO CD8 T cells are actually better killers.*

-We agree with reviewer's concern and to address this question we designed and performed a competitive *in vivo* killing assay. In support of our hypothesis that *Grail*^{-/-} CD8⁺ T cells are better killers, this new set of data demonstrates that *in vivo* primed antigen-specific *Grail*^{-/-} CD8⁺ T cells have better killing ability towards antigen-pulsed target cells. Overall this new data supports the function of Grail in controlling CTL cytotoxic activity and has been added as **Figure 3f**.

Fig. 2: The CD8 transfer experiments demonstrate much better tumor elimination in vivo with the Grail KO cells, but the CD4 and CD8 accumulation in the tumor are not greatly increased. In Fig 2e, there are actually more Gzmb⁺ cells in the WT than in the Grail KO. Again, are the Grail KO cells really better killers in vivo or is this an indirect effect?

-In **Fig. 2c and d** the percentage of total CD4⁺ and CD8⁺ TILs were analyzed. We determined a marked increase in the percentage of total, donor, and host CD8⁺ T cells in TILs from *Grail*^{-/-} OT-I-transferred group compared to control and OT-I transferred group. We would like to emphasize that in all our experiments, we consistently saw significant impact of Grail on double positive Gzmb and IFN γ producing CD8⁺ T cells but not Gzmb single positive CD8⁺ T cells. Thus, we conclude that increased percentage of double positive Gzmb/IFN γ producing CD8⁺ TILs in *Grail*^{-/-} mice is responsible for tumor killing. To assess *Grail*^{-/-} CD8⁺ T cells killing ability, we have performed an *in vivo* killing assay and determined a significant killing ability of antigen-specific *Grail*^{-/-} CD8⁺ T cells compared to WT CD8⁺ T cells. The text has been clarified to demonstrate that the major impact of Grail is on the production of IFN γ and not Gzmb.

Fig. 3: In the bone marrow chimera experiments, again there are more Gzmb⁺ cells in the WT than in the Grail KO. The effect on Gzmb⁺Ifng⁺ CD8 is very small. Fig 3c-e show effects on naïve sorted CD8 T cells in vitro. The induction of Gzmb⁺Ifng⁺ cells is quite substantial and the cytolytic activity against OT1 pulsed EL-4 is higher, but this in vitro assay does not necessarily translate to an in vivo killing effect in the tumor.

-As indicated above, we have detected a statistically significant increase in the percentage of double positive Gzmb/IFN γ producing CD8⁺ TILs in the absence of Grail. In addition, enhanced effector function of *Grail*^{-/-} CD8⁺ T cells was confirmed by the *in vivo* killing assay shown in **Figure 3f**. The text has been clarified to state that the impact of Grail is on IFN γ production by Gzmb⁺ CD8⁺ T cells.

Fig. 4: It is not clear why the role of IL-21 in Grail inhibition of CD8 killing was investigated. Fig 4a-c document in vitro effects of IL21 on WT and Grail KO CD8 T cells, but this is not sufficient to connect the Grail effect to the IL-21 effect. In the tumor experiment shown in Fig. 4d, although lack of IL-21 does away with the enhanced tumor killing in the Grail KO, the tumor curve for the IL-21 KO is not included-is it perhaps worse than the WT, in which case the IL-21 effect and the Grail effect are distinct.

-We apologize for the confusion. We attempted to show the role of Grail in controlling IL-21R expression and consequently in regulation of IL-21R-signalling in CD8⁺ T cells. We found that *Grail*^{-/-} CD8⁺ TIL expressed significant levels of IL-21R compared to WT CD8⁺ TIL. This was the reason why we assessed whether elevated IL-21R expression and IL-21R signaling in *Grail*^{-/-} CD8⁺ T cells could account for their enhanced effector functions *in vitro* and *in vivo* towards tumor growth. In fact, we found that naïve *Grail*^{-/-} T cells are more sensitive to IL-21 treatment compared to WT CD8⁺ T cells *in vitro*. We have also confirmed the importance of IL-21 signaling in the function of *Grail*^{-/-} CD8⁺ T cells *in vivo*.

We agree that IL-21 KO group has to be included to show the impact of IL-21 signaling on function on WT CD8⁺ T cells in the lymphoma model. In the revised manuscript, we have compared the tumor growth between WT and IL-21 KO mice and did not find significant difference between groups (**Supplementary Figure 8**). In addition, the percentage and effector function of CD8⁺ TILs were comparable between the two groups. The lack of difference observed between the IL-21 KO and WT groups could be explained by the high expression level of Grail in WT CD8⁺ TILs being sufficient to suppress IL-21R expression and consequently preclude IL-21 signaling in CD8⁺ T cells. This data and supporting text has been added to the revised manuscript.

Fig. 5: Original flow data should be displayed (5c)-it is unlikely that naïve CD8 T cells express high levels of IL-21R. The ubiquitination of IL-21R components in 293T cells should not be interpreted as proof of a role of ubiquitination in CD8 T cells.

-As suggested above, in the revised manuscript we have incorporated flow data for IL-21R α and IL-2R γ expression for naïve WT and *Grail*^{-/-} CD8⁺ T cells in **Fig. 5C**. This data suggests that naïve *Grail*^{-/-} CD8⁺ T cells express higher levels of IL-21R. To determine whether IL-21R is a direct target, we performed *in vitro* ubiquitination assay by overexpressing IL-21R, WT Grail, Grail mutant and HA-Ub in 293 T cells. We found that WT Grail targets for ubiquitination both IL-21R α and IL-2 γ chains. We agree with the reviewer that *in vivo* ubiquitination assay will be the best to confirm the function of Grail towards IL-21R expression in CD8⁺ T cells; however, we could not perform this assay due to lack of a commercially available IL-21R antibody for immunoprecipitation of the IL-21R α chain in CD8⁺ T cells. Nevertheless, an alternate approach to address this point is to determine whether Grail regulates the expression of IL21R α and IL-2R γ through endosome-mediated degradation. To test this, naïve OT-I T cells from WT mice were activated with OVA peptide and WT APCs in the presence of the endosome inhibitor Latrunculin B. Addition of Latrunculin B led to upregulation of IL-21R α and IL-2R γ expression, suggesting that Grail targets endocytosed IL-21R complex via endosome-mediated degradation in CD8⁺ T cells. This additional data has been added as **Figure 5e**.

Fig. 6: Although TGF β clearly upregulated Grail protein in activated CD8 T cells in vitro, blocking TGF β in vivo was not shown to affect Grail protein expression in TILs, nor was it shown to impact tumor growth.

-As suggested by the reviewer, in our revised manuscript we have added data demonstrating that administration of blocking antibodies to TGF- β impacts tumor growth and expression of Grail but not Cbl-b in TILs. This data is found in **Figure 6b and c**.

Fig. 7: It is enticing to believe that enhanced Grail expression in the patient CD8⁺ T cells is a result of higher TGF β levels and that it leads to reduced IL-21R levels, but no causal relationship is demonstrated here.

-The main focus of this manuscript is to determine the novel function of Grail in CD8⁺ T cells using mouse models. By screening of lymphoma patients, we found the similar correlation as seen in the mice between high expression of Grail and diminished expression of IL-21R in CD8⁺ T cells compared to CD8⁺ T cells from healthy donors. Moreover, we suggest this could be due to high levels of TGF- β that we (and others) have found to be present in patient serum. We agree with the reviewer that it will be important to show a mechanism for Grail function in patient CD8⁺ T cells; however, it is beyond the scope of the current manuscript. In the future, we plan to focus more on analysis of human samples and

determine the role of Grail in the function of CD8⁺ T cells in order to develop a pre-clinical approach for targeting Grail in patient TILs. The serum data has been added as **Figure 7e**.

Reviewer #2

This work is based on the knowledge that the E3 ubiquitin ligases Itch, Cbl-b and Grail may be involved in the regulation of T cell function. The authors examined Grail in murine and human CD8 T cells. They used EL4 and EG-7 cells as transplantable tumor model and also briefly studied CD8 T cells from lymphoma patients. Higher levels of Grail were found to be associated with lower levels of IL-21R in both species. The mouse experiments further show a role for TGFb in the upregulation of Grail and reduced IL-21R expression. In absence of Grail, CD8 T cells produced more IFNg and GrzmB and conferred better antitumor effects in vivo.

The study is based on a meaningful hypothesis. However, the data is somewhat unclear, only based on a transplantable tumor model, and very limited results from patient's PBMC.

-We agree that in the current format for the most of our studies, we utilized the mouse system. Analysis of patient samples has allowed us to determine whether there is a correlation between Grail expression and function in mouse and human CD8⁺ T cells. We agree that further studies are essential to determine the exact function of Grail in human T cells; however, it is beyond the scope of the current manuscript.

Increased GzmB production in absence of Grail does not seem to be very consistent, when focusing on Figure 2e that shows that Grail-/- OT-1 cells produced more IFNg but not more GrzmB than OT-1 cells. The same applies for Figure 3c, right side. These results are different as compared to the data shown in Figure 1d and other parts of Figure 3. Nevertheless, the authors state that the data shown are representative.

- As we indicated in the answer to the first reviewer, in all our experiments, we consistently detected a significant impact of Grail on double positive GzmB and IFN γ producing CD8⁺ T cells but not GzmB single positive CD8⁺ T cells. Thus, we conclude that the increased percentage of double positive GzmB/IFN γ producing CD8⁺ TILs in *Grail*^{-/-} mice is responsible for tumor killing. This has been clarified in the revised manuscript text.

In the dot plots of Figure 3a, the gating and the percentages of GzmB +/- cells is not shown, which is inconsistent to the remaining data on this subject. On several occasions, it is difficult to explain what the authors mean with "representative data" when comparing the dot plots with the histograms. I suggest that the authors provide all the values of all experiments, which can also be done e.g. as tables and/or as supplementary material.

-We apologize for any confusion. We did not compare dot plots with bar-graphs. The dot plots contain representative data from one group. The bar-graphs contain individual mouse data points as well as the mean \pm SEM for each group. In the revised manuscript, we clarified the figure legends and included the independent data points for the experiments.

Supplementary Figure 5 only shows a single dot plot, without x- and y-axis labels, and there is no a) and b) in this Figure. Thus, it is not possible to review the data on TCR expression.

-**Supplementary Figure 7a** was corrected in revised manuscript to include the appropriate labeling.

In the first part, the authors state that Grail has no effects on CD8 T cell proliferation. However, subsequently (Figure 4 and text), the data show different proliferation and the authors conclude on "expansion".

-In **Figure 4b**, we determined that there was a difference in proliferation between WT and *Grail*^{-/-} CD8⁺ T cells in their response to IL-21. In the rest of the experiments, we did not see the difference in proliferation between groups in the response to TCR signaling and costimulation through either anti-CD28 or exogenous cytokines other than IL-21. This has been clarified in the text.

The authors repetitively state that "Grail-deficient CD8+ T cells are sufficient to (completely) reject established tumors". However, the majority of data show that tumors were not completely rejected.

-We agree with reviewer's comment and in the revised manuscript, we have clarified the text to state that *Grail*^{-/-} CD8⁺ T cells are sufficient to control growth of established tumors.

Also at several other occasions, "greatly increased" or similar wording does not reflect the often relatively small differences, suggesting over-interpretation of the findings. Furthermore, the multiple roles of Grail, beyond the ones studied here, are insufficiently investigated.

-In the revised manuscript, the significant difference between analyzed groups was assessed based on statistical analysis. The role of Grail in CD8⁺ T cells is novel and has not been previously studied. In the current manuscript we focus on the role of Grail towards TCR signaling and IL-21R signaling; however, we cannot disregard that additional mechanisms potentially exist and will be studied in the future.

The data provided from lymphoma patients is promising but very limited. Additional experiments assessing T cell function are necessary in order to justify that the author's conclusion also apply to humans.

-As stated in the above responses, we agree that further studies are essential to determine the exact function of Grail in human CD8⁺ T cells: however, it is beyond the scope of the current manuscript.

Minor point: "TILs" may be replaced with "T cell" in the first subtitle of the results section ("Grail loss correlates with enhanced CD8+ TILs infiltration in lymphoma tumors")

-The TIL terminology we are using is for characterization of immune cells from the tumor; however, for some experiments we analyzed CD8⁺ T cells from mice that were not exposed to tumors. As such, the text uses either TIL or T cells according to the site of examination.

Reviewer #3

The manuscript by Wang et al investigates the role of Grail in anti-tumour immunity mediated by CD8+ T cells. While their observations are potentially interesting, the quality of the work is a concern. The major issues of concern are listed below:

The majority of the experiments were only performed twice and, almost invariably, the authors show only representative data. In some instances, it appears as though the experiments were only done once, e.g. Fig. 5A, B, D, while the number of replicates in

experiment C was unclear. In another example, the authors state in the legend for Figure 4 that "the expression level in WT cells was set as 1". Where does the error bar in WT samples come from then? There are many other uncertainties of a similar kind found in most of the figures.

-We seriously took into consideration the reviewer's concern and revised the figures and figure legends accordingly to address the above points.

The experimental protocols are inconsistent. For example, in Figure 3, the T cells were activated for different periods of time: 3 days (3C), 2 days (3D) and 7 days (3E). In a similar experiment shown in Suppl Fig. 6B, the cells were activated for 24hrs. Given the disparity in the activation periods between experiments, the interpretation of these results must be questioned.

-We used different time points for CD8⁺ T cell activation *in vitro* to detect protein levels of IFN γ and GzmB. Based upon the type of TCR signaling induced, we activated CD8⁺ T cells for 3 days (with CD3/CD28) or 2 days (with peptide). Only 24h hours of activation of T cells was needed to determine mRNA expression levels of the indicated genes. The methods and other appropriate text have been clarified to address this point.

I also question the design of the in vivo experiments, where all the mice were culled on Day 17 - the day that WT mice had to be culled due to high tumour burden. At that point, Grail^{-/-} animals could still control (but not reject) the tumours. These mice should be observed for at least another 5-7 weeks to demonstrate the prolonged/constitutive anti-tumour activity of Grail^{-/-} T cells (or tumour rejection). As shown by the authors, tumours can "overwhelm" the animals within less than a week. Therefore, the data presented does not provide sufficient information to justify the conclusion of this part of the study.

-We thank reviewer for the good suggestion. To address this point, we performed long-term analysis of Grail^{-/-} mice after tumor inoculation. Consistent with our previous experiments, WT mice were euthanized on day 17 due to high tumor burden compared to Grail^{-/-} mice. Grail^{-/-} mice were observed for an additional 2 weeks with the same tumor pattern, suggesting the role of Grail^{-/-} TIL in controlling tumor growth rather than in complete rejection. This data has been added to the revised manuscript in **Supplementary Figure 2**.

TILs were only analyzed on Day 17, when WT mice had to be culled due to uncontrolled tumour growth. Had the TILs been assessed earlier, when the tumours in WT and Grail^{-/-} animals were still small and remained under the putative control of the immune system - e.g. on Days 9-11 - it would perhaps be possible to determine a mechanism that allowed Grail^{-/-} mice to control tumour growth. For the authors to interpret this data adequately, and draw solid conclusions, the TILS would need to analyzed at various time points during the course of tumour growth, particularly during the critical early period of immune control.

-This is an excellent point that was raised by the first reviewer as well. To address this point, we performed a time course TIL analysis in WT and Grail^{-/-} mice. We did not detect differences in the percentage of tumor-infiltrating CD4⁺ and CD8⁺ T cells between WT and Grail^{-/-} groups on day 7 day after tumor inoculation; however, on day 12 we observed that the percentage of CD8⁺ TIL was significantly increased in Grail^{-/-} mice compared to WT counterparts. To explore the possible mechanism(s) for accumulation of Grail^{-/-} CD8⁺ T cells in the lymphoma, first we analyzed the susceptibility of WT and Grail^{-/-} CD8⁺ TILs to apoptosis; however, we did not detect any difference in the percentage of apoptotic markers Annexin V and 7-AAD in CD8⁺ TIL between WT and Grail^{-/-} groups on either day 7 and 12

after tumor inoculation. Next, we analyzed the expression pro-survival costimulatory receptors OX-40 and 4-1BB and anti-apoptotic factor Bcl-2 in CD8⁺ WT and *Grail*^{-/-} TILs on days 7 and 12 after tumor inoculation. Interestingly, we found increased expression of OX-40, 4-1BB and Bcl-2 by *Grail*^{-/-} CD8⁺ TILs on both days. Moreover, mRNA levels of pro-survival costimulatory molecules OX-40 and 4-1BB and anti-apoptotic gene Bcl-xL (B-cell lymphoma-extra large) were enhanced on day 17 as well, suggesting that Grail could control the survival of CTLs infiltrated into the tumors. This data has been added to the revised figures (**Fig. 1b and Supplementary Fig. 4a**) and the text clarified accordingly.

The experimental procedures are also poorly described. For example, anti-CD3/CD28 or SIINFEKL stimulation of the T cells had to be done in the presence of IL-2 (this was not stated in the M&M). If so, how can the results in Suppl Fig 6B (how many times was this experiment done?) showing the identical IFN-gamma mRNA levels following IL-2 treatment (concentration is unknown) be reconciled with a large difference in the intracellular levels of IFN-gamma between WT and Grail^{-/-} shown in Figure 3C,D. I understand that these are two different readouts (although done on different days), but if so, what is the biological relevance of mRNA levels shown in Figures 1, 4, 6, 7? This discrepancy should be addressed in the manuscript.

-Experimental procedures were revised and clarified to include a detailed experimental set-up. IL-2 treatment was only used in **Supplementary Figure 9b and c** in order to assess the impact of Grail in IL-2 signaling. In the rest of the *in vitro* experiments, we did not use IL-2. Explanation for the difference in experimental set-up and biological relevance of mRNA expression has been incorporated into the revised manuscript.

Reviewer #4

Whereas the role that Grail has as a regulator of CD4 T cell activation has been well characterized, little was known about whether or how this E3 ubiquitin ligase might control CD8 T cell function. In this study using a model of s.c. transfer of lymphoma, the authors identify Grail as a regulator of the anti-tumor CTL response. The data show that CD8 TILs upregulate the expression of Grail where it likely dampens effector functions and responses to IL-21, as CD8 T cells deficient in Grail show increased IFN γ expression and increase response to IL-21 that results in enhanced control of tumor growth. The paper identifies, thus, a novel and very relevant regulator of anti-tumor CD8 T cell responses and should help us understand better the mechanisms that mediate tumor-induced immune evasion.

There are a series of points that would need to be addressed, though, to fully support some of the conclusions reached by the authors and better understand how Grail controls CTL function.

Base on the data presented, the conclusion reached by the authors is that Grail is not involved in the direct regulation of CD8 T cell activation but specifically in the regulation of effector cytokine expression by those cells. The data supports for the most part this assertion but there are a few points that need clarification. While all the FACS data is described as changes in the number of GrzB⁺IFN γ ⁺ CD8⁺ T cells, many FACS analyses show little if any difference in the expression of GrzB (e.g. Figs 2c-e; Fig 3 b-d...) and the increased in the % of double positive cells in those experiments reflects merely and increased in the cells

expressing of IFN γ . Would IFN γ be the more specific target of Grail regulation rather than both IFN γ and GrzB? Aside for the studies on IL-21R signaling (see comments below) one possible proposed mechanism that may explain the role of grail is its ability to modulate TCR downregulation. If this is true, how would the authors explain those changes inavailability of TCR can affect the expression of some cytokines (i.e. IFN γ) and not others (i.e. IL-2)?

-We agree with the reviewer's comment that the main impact of Grail is on IFN γ and not GzmB production. The text has been clarified to this point. The reviewer makes an interesting observation about the selective increase in IFN γ expression. There are a couple of possibilities to explain this point. First, IL-21 signaling has been demonstrated to promote IFN γ secretion. This is cited in the text. Additionally, the strength of TCR signaling has been shown to be important in the production of IFN γ . It is possible that there is a synergistic effect of the enhanced TCR expression and enhanced IL-21R signaling that results in the overall enhanced IFN γ production observed in *Grail*^{-/-} CD8⁺ T cells.

The other proposed target of Grail is the IL-21R. The authors show increased sensitivity of CD8⁺ T cells that lack Grail to IL-21 and loss of the ability of Grail^{-/-} mice to control EG7 growth in IL-21 deficient mice. The authors proposed that Grail-mediated ubiquitination of the common gamma chain and the IL-21Ralpha chain would regulate their degradation, which would account for the increased expression of the IL-21R in Grail deficient CTLs. However, the paper does not show any experiment to provide evidence of this increased turnover. This is further complicated by the fact that there is a decreased expression of the mRNAs for both genes. Does Grail also control gene expression of the IL21R subunits or are those changes a result of some positive feedback? The data on ubiquitination in the 293 system is also a little difficult to interpret. Multiple bands that do not appear to correlate with multiples of Ub appear on the IPs which may suggest some problems with the IP in an overexpression system. It would help if it could be shown for instance that in this system, one a specific chain of one of the other common gamma chain cytokine receptor that do not appear to be affected by Grail, is indeed not ubiquitinated when Grail is overexpressed.

-We found that naïve *Grail*^{-/-} CD8⁺ T cells express higher levels of IL-21R compared to WT CD8⁺ T cells indicating that Grail potentially targets IL-21R for degradation. In fact, our *in vitro* ubiquitination assay confirmed our hypothesis. After immunoprecipitation of IL-21R subunits and probing with HA-HRP antibodies, we observed the formation of high molecular weight ubiquitin conjugates in the presence of WT Grail, indicating a polyubiquitin chain formation (this is the reason for the multiple bands seen on the western blot analysis). However, this ubiquitination effect was greatly diminished in the presence of the Grail mutant. Thus, IL-21R is a potential substrate for Grail-mediated ubiquitination. To determine whether Grail regulates the expression of IL-21R through endosome-mediated degradation, naïve OT-I T cells from WT mice were activated with OVA peptide and WT APCs in the presence of the endosome inhibitor Latrunculin B. Addition of Latrunculin B led to upregulation of IL-21R α and IL-2R γ expression, suggesting that Grail targets endocytosed IL-21R complex via endosome-mediated degradation in CD8⁺ T cells. This supporting data has been added to **Figure 5e**.

Based on the reviewer's suggestion, we also checked specificity of Grail towards IL-21R subunits by employing an IL-15R α *in vitro* ubiquitination assay. We did not detect formation of any ubiquitin conjugates following IL-15R α immunoprecipitation compared to immunoprecipitation of IL-2R γ , further supporting that Grail specifically controls expression of IL-21R through an ubiquitin-dependent endosome degradation pathway. The additional

data has been added to the revised manuscript (**Supplementary Figure 10**) and the text clarified to address this point.

The experiments that identify TGF β as a possible regulator of Grail expression in CTLs are very interesting as they could help explain how the expression of this protein is upregulated in a tumor microenvironment frequently rich in TGF β . It would be interesting, though, to verify that the increased in Grail caused by TGF β is enough to downmodulate the expression of the IL-21R in those cells.

-As suggested by the reviewer, in our revised manuscript we have added data demonstrating how administration of blocking antibodies to TGF- β impacts the expression of Grail and IL-21R in TIL. This data has been added to **Figure 6c and d**.

Other points:

Representative FACS images for Fig. 5c should be shown.

- We have included FACS images for **Figure 5c** in our revised manuscript.

In several instances it is claimed that transfer of Grail $^{-/-}$ CD8 $^{+}$ T cells is sufficient to mediate rejection of established EG-7 tumors to support a possible therapeutic value. However, in those experiments cells were transfer to animals on day 5 when tumors are still not detectable. It might be a good idea to more precisely define those experiment or alternatively analyze the possible effect of CTL transfer when tumors are already detectable (day 9-10?)

-We have clarified in the revised text that CD8 $^{+}$ T cells were injected into tumor-bearing mice on day 5 after tumor inoculation. At this time point the tumor was palpable.

In Fig 3b, it would be interesting to complete the data by showing any differences in the % of CD45.1 and CD45.2 CTLs that infiltrate the tumors.

-The percentage of donor and recipient CTLs infiltrated into the tumors has been added to **Fig. 2d** of the revised manuscript.

Regarding the differences in expression of mRNAs for cytokines in sorted TILs shown in Fig. 1e, though they appear statistically significant, it is difficult to clearly asses a biological meaning to increases in gene expression that seem to range from 30 to 60% (all less than 2-fold). This is likely the result of the limited % of T cells that might be expressing those cytokine genes at a given time in vivo, but I am not sure the data as it is can support biological relevance of those increases.

-We agree with the reviewer that mRNA expression pattern differences between the WT and Grail $^{-/-}$ T cells are not above 2 fold; however, they are still statistically significant. Nonetheless, the key point is the differential protein expression between the CD8 $^{+}$ T cells from WT vs Grail $^{-/-}$ mice. Indeed, the data demonstrates an increased IFN γ^{+} population that co-expresses GzmB in the Grail $^{-/-}$ T cells (**Figure 1e**). This has been also clarified in the text and in other figures showing IFN γ and GzmB expression. In addition, we have added biological data demonstrating the differential killing capabilities of these subsets *in vivo* (**Figure 3f**).

Reviewers' comments:

Reviewer #1 (Remarks to the Author):

Although the authors have addressed some of my previous concerns, the relationship between Grail and IL-21R is still not clear. It is still not clear that Grail deficiency causes hyper-responsiveness of CD8+ T cells to IL-21R signaling. In Figure 4b, the authors conclude that the Grail KO CD8 cells have enhanced responses to IL-21, but in fact, the baseline responses (no IL-21) of the KO mice for all of these responses are higher, suggesting other factors (developmental?) as potentially responsible.

The requested tumor curve for the IL-21KO mice, although included in the Supplement, is not part of the experiment in Fig. 4d and it is therefore not possible to directly compare all 4 strains of mice.

In Figure 5C, the level of IL-21R on Grail KO CD8 T cells is not significantly higher than that on WT CD8 T cells. Are the isotype controls specific to each of these populations?

The experiment addressing whether Grail regulates IL21R post-translationally (Figure 5d) is still done using 293T cells, and this effect in an overexpression system cannot be inferred to be true for CD8+ T cells. The lack of an effect on IL-15Ra in CD8 cells does not address this problem. Results using the endosome inhibitor latrunculin B in Fig 5e may be an artifact—are there any proteins that are not upregulated on the surface by drug treatment?

Finally, the overexpression of Grail in patient lymphomas has not been directly related to the reduced IL-21R expression in these patient cells. Many other mechanisms are possible.

Reviewer #2 (Remarks to the Author):

Thank you for the revision. The manuscript has improved, the authors have replied productively to my comments.

A detail: I still wonder about the wording "enhanced CD8+ tumor infiltrating T cells infiltration" (first subtitle of results section, using the abbreviation TILs)

Reviewer #3 (Remarks to the Author):

Wang et al have addressed some of my concerns and included the new data to strengthen their argument.

The results of a longer-term tumour growth experiment (Suppl Fig 2) suggested cancer immune-equilibrium in Grail^{-/-} mice. However, it is unclear how this "equilibrium" will be resolved, ie will the tumour escape immune surveillance, will it be rejected or will it persist? To resolve this, I had suggested observing the mice for additional 5-7wks. What I find puzzling is that the authors extended their experiment only by 2 weeks, when presumably this matter could have been studied in the same experiment. They also neglected to analyse TILs after 31days (to compare, for example with day 17). Furthermore, this was the only one out of several in vivo experiments, where they extended their observation of tumour-bearing mice. Why?

What role do Grail^{-/-} T regs play in tumour rejection? An earlier paper by Nurieva et al (Immunity, 2010) strongly suggested downregulation of Treg suppressive function.

New Figure 3f shows a complete elimination of target cells by Grail^{-/-} CD8⁺ in in vivo cytotoxicity assay (surprisingly, wild-type CD8⁺ showed a very weak killing potential). An in vitro cytotoxicity assay (eg 51Cr release) showing that Grail^{-/-} CD8 T cells were considerably more potent than WT cells should easily be achievable given the OT-1 background of the experimental animals. An assay shown in Figure 3e is inadequate.

Looking at the very mild increase in granzymeB and perforin levels, I am not convinced that this would have a significant effect on T cell function. It is theoretically possible, but needs to be validated. Even a simple in vitro transduction of wild-type T cells with GzmB or perforin and an in vivo cytotoxicity assay will help to rule in or out the effect of cytotoxic molecules. Why some of the tests were conducted using FACS and others using gene transcription? As a minor, but important point, the data there was normalised against WT (eg Fig 4) thus eliminating any sample-to-sample variation for WT and artificially inflating statistical differences; instead, paired or unpaired t-test should be used, without normalisation.

Is TCR "downmodulation" shown in Suppl Fig 7a due to its internalisation or degradation? The Western blot in Suppl Fig 7B is poor in quality and difficult to interpret (especially compared to their own data published in Immunity 2010, Figure 5D).

Overall, while the data in this paper is potentially interesting, a lot more work needs to be done to validate their observations and provide a mechanism for anti-tumour activity of Grail^{-/-} CD8⁺.

Reviewer #4 (Remarks to the Author):

The authors have revised this manuscript addressing all concerns raised in the original submission. The text has been modified to more clearly describe the role of Grail as a regulator of IFN γ expression and anti-tumor effector function in CTLs and new data has been included to support a specific role of Grail in the regulation of IL-21R turnover and signaling, better define the effect of TGF β on Grail expression and to better characterized the functional consequences of the loss of Grail in CTLs.

I just want to point a couple of minor details the authors may want to address. Fig.2d shows increased infiltration not only of donor Grail-deficient CD8⁺ T cells but also of host Grail-sufficient CTLs when OT-I Grail^{-/-} CD8⁺ T cells are transferred into tumor-bearing mice. This could be explained in the line suggested by one of the reviewers that maybe increased tumor death induced by the rail-deficient CD8⁺ T cells may lead to increased infiltration of not only donor but also host T cells. It would interesting if the authors could speculate in the discussion possible explanations for that effect. Finally, the legend of Suppl. Fig 1 indicates the experiments were performed twice but not how many mice were analyzed per experiment.

POINT-BY-POINT RESPONSE TO REVIEWERS' COMMENTS

We thank all reviewers for their critiques, comments and helpful advice. We have provided additional textual clarification as well as experimental evidence of the role of Grail in CD8⁺ CTL function, mediation of tumor control, and specific role in the regulation of IL-21 receptor signaling. We hope these edits will address all the critiques satisfactorily. All changes in the text are underlined.

Reviewer #1:

1. Although the authors have addressed some of my previous concerns, the relationship between Grail and IL-21R is still not clear. It is still not clear that Grail deficiency causes hyper-responsiveness of CD8⁺ T cells to IL-21R signaling. In Figure 4b, the authors conclude that the Grail KO CD8 cells have enhanced responses to IL-21, but in fact, the baseline responses (no IL-21) of the KO mice for all of these responses are higher, suggesting other factors (developmental?) as potentially responsible.

-We agree with the reviewer that other mechanisms are possible. An interesting possibility is that due to the degradation of the TCR by Grail following the anti-CD3 stimulation, the level of IL-21R may be impacted as its expression is a result of TCR triggering. To test this, (**new Figure 4e**), WT and Grail KO CD8⁺ T cells were stimulated with anti-CD3 with and without IL-21. Changes in IL-21R expression was determined using flow cytometry and demonstrates that Grail KO CD8⁺ T cells have enhanced IL-21R expression in both conditions as compared to the WT counterparts. The addition of IL-21 to the stimulation further enhanced the level of IL-21R expression over that observed with a TCR stimulation alone. We also demonstrated that the induction of IFN γ and GzmB following a TCR stimulation is also enhanced in the Grail KO CD8⁺ T cells at the protein level by flow cytometry in support of our ELISA and gene expression data also shown in **Figure 4c-d**. Like the expression of IL-21R, the addition of IL-21 to the stimulation condition further enhanced the expression of these cytotoxic markers. This data also supports the hypothesis that the improved functionality of Grail KO CD8⁺ T cells is not only due to proliferation as flow cytometry demonstrates changes at the per cell level. Thus, we propose that absence of Grail in CD8⁺ T cells leads to more cytokine production at both the per cell and population levels.

2. The requested tumor curve for the IL-21KO mice, although included in the Supplement, is not part of the experiment in Fig. 4d and it is therefore not possible to directly compare all 4 strains of mice.

-We agree with the reviewer that ideally all strains would be included in the experimental repeat. However, we did not have enough mice to include all the strains and so concluded that comparing to the WT would be scientifically sound in this case. This conclusion was reached after we determined that there was no difference in tumor growth between the WT and IL-21KO mice.

3. In Figure 5C, the level of IL-21R on Grail KO CD8 T cells is not significantly higher than that on WT CD8 T cells. Are the isotype controls specific to each of these populations?

-This experiment uses naïve T cells as the biological control as opposed to the use of an isotype control for the flow cytometry. However, we have also added more data showing the expression of IL-21R on CD8⁺ T cells activated with anti-CD3 in the presence or absence of IL-21 from both WT and KO mice. This data has been added to **Figure 4e**.

4. The experiment addressing whether Grail regulates IL21R post-translationally (Figure 5d) is still done using 293T cells, and this effect in an overexpression system cannot be inferred to be true for CD8+ T cells. The lack of an effect on IL-15Ra in CD8 cells does not address this problem. Results using the endosome inhibitor latrunculin B in Fig 5e may be an artifact—are there any proteins that are not upregulated on the surface by drug treatment?

-We agree with the reviewer that this is an artificial system and performed the experiment to show that not all proteins are upregulated by drug treatment. To this end, T cells were treated with lantrunculin B and assessed for expression of IL-7R, IL-2Ra (CD25), and IL-2Rb (IL-15Ra, CD122). This experiment demonstrates that these proteins, in fact, are not modulated by drug treatment. This data has been added to **Figure 5e** and the results section and legend updated accordingly.

5. Finally, the overexpression of Grail in patient lymphomas has not been directly related to the reduced IL-21R expression in these patient cells. Many other mechanisms are possible.

[Redacted]

Reviewer #2:

Thank you for the revision. The manuscript has improved, the authors have replied productively to my comments.

-We would like to thank Reviewer 2 for their time and effort in providing valuable critiques to the manuscript.

Reviewer #3

*1. The results of a longer-term tumour growth experiment (Suppl Fig 2) suggested cancer immune-equilibrium in *Grail*^{-/-} mice. However, it is unclear how this “equilibrium” will be resolved, ie will the tumour escape immune surveillance, will it be rejected or will it persist? To resolve this, I had suggested observing the mice for additional 5-7wks. What I find puzzling is that the authors extended their experiment only by 2 weeks, when presumably this matter could have been studied in the same experiment. They also neglected to analyse TILs after 31days (to compare, for example with day 17). Furthermore, this was the only one out of several *in vivo* experiments, where they extended their observation of tumour-bearing mice. Why?*

-The reviewer is correct in that a longer timeline was suggested in the initial review. As such, this experiment was repeated and extended for a total of 55 days. In fact, while an equilibrium was observed in *Grail*^{-/-} mice, eventually tumors were found to begin to grow albeit slowly throughout the time of the experiment. As such, we believe this supports the role of *Grail* in CD8⁺ T cell anti-tumor immunity but demonstrates the presence of acquired resistance. In addition, we have provided CD4 and CD8 staining as well as GzmB and IFN γ in *Grail*^{-/-} TIL from day 55 following tumor inoculation. This data has been added to **Supplementary Figure 2a and b**.

*2. What role do *Grail*^{-/-} T regs play in tumour rejection? An earlier paper by Nurieva et al (Immunity, 2010) strongly suggested downregulation of Treg suppressive function.*

-We agree that Tregs could play a role in tumor rejection since *Grail* absence could also impact this T cell subset. As such, *Grail*^{-/-} Tregs were assessed for their suppressive function *in vitro* (**Supplementary Figure 4a**). In fact, *Grail*^{-/-} Tregs were found to be less suppressive of both WT and *Grail*^{-/-} CD8⁺ T cells. Importantly, *Grail*^{-/-} CD8⁺ T cells were less susceptible to suppression by WT Tregs demonstrating an additional mechanism for enhanced function of *Grail*^{-/-} CD8⁺ T cell within the tumor microenvironment. Overall this data supports that the combination of *Grail*^{-/-} CD8⁺ T cell resistance to Treg mediated suppression along with the reduced ability of *Grail*^{-/-} Tregs to suppress CD8⁺ T cells results in enhanced tumor control. This data has been added as **Supplementary Figure 4a** and the text has been updated accordingly.

*3. New Figure 3f shows a complete elimination of target cells by *Grail*^{-/-} CD8⁺ in *in vivo* cytotoxicity assay (surprisingly, wild-type CD8⁺ showed a very weak killing potential). An *in**

in vitro cytotoxicity assay (eg 51Cr release) showing that *Grail*^{-/-} CD8 T cells were considerably more potent than WT cells should easily be achievable given the OT-1 background of the experimental animals. An assay shown in Figure 3e is inadequate.

- For the *in vivo* killing assay, we are also surprised at how quickly the *Grail*^{-/-} CD8⁺ T cells kill their target. However we would like to stress that the WT T cells are still able to kill their target *in vivo*; this just takes a few more hours than the *Grail*^{-/-} T cells. To address this point, the *in vivo* killing assay was extended 8 hours to assess the CTL function of WT CD8⁺ T cells. This additional data has been added to **Supplementary Figure 7**.

4. *Looking at the very mild increase in granzymeB and perforin levels, I am not convinced that this would have a significant effect on T cell function. It is theoretically possible, but needs to be validated. Even a simple in vitro transduction of wild-type T cells with GzmB or perforin and an in vivo cytotoxicity assay will help to rule in or out the effect of cytotoxic molecules.*

- In terms of *in vitro* transduction of T cells with GzmB or perforin, we are not saying that the WT T cells are unable to kill or to produce GzmB or perforin. The data are just showing that the absence of *Grail* in CD8⁺ T cells results in better killing capabilities than the WT counterparts. It is unclear to us what overexpression of these cytolytic molecules will tell us about the role of *Grail*. In fact, an *in vivo* cytotoxicity assay was performed and demonstrates that *Grail* KO CD8⁺ T cells are able to kill better than their WT counterparts.

5. *Why some of the tests were conducted using FACS and others using gene transcription? As a minor, but important point, the data there was normalised against WT (eg Fig 4) thus eliminating any sample-to-sample variation for WT and artificially inflating statistical differences; instead, paired or unpaired t-test should be used, without normalisation.*

- We are happy to show the data and have added t-tests for statistical significance using the raw data without normalization against the WT. The flow cytometry versus gene transcription is simply to determine if the differences observed are at both the protein and RNA level, which they are.

6. *Is TCR “downmodulation” shown in Suppl Fig 7a due to its internalisation or degradation? The Western blot in Suppl Fig 7B is poor in quality and difficult to interpret (especially compared to their own data published in Immunity 2010, Figure 5D).*

-The TCR downmodulation in the text refers to its degradation not internalization. Indeed, a western blot would show no difference if the TCR was only internalized as the overall protein level would not change. We have modified the text for clarity. In terms of the western blot being poor in quality, perhaps this is due to the quality of the image in the compiled pdf. If so we apologize as the quality of the western blot itself is very comparable (if not better) than that previously published in *Immunity*.

Overall, while the data in this paper is potentially interesting, a lot more work needs to be done to validate their observations and provide a mechanism for anti-tumour activity of Grail-/- CD8+.

-We hope that the addition of textual clarification as well as additional experimental evidence has convinced the reviewer of the role of Grail in CD8⁺ CTL function that can be applied to a tumor setting.

Reviewer #4

1. The authors have revised this manuscript addressing all concerns raised in the original submission. The text has been modified to more clearly describe the role of Grail as a regulator of IFN γ expression and anti-tumor effector function in CTLs and new data has been included to support a specific role of Grail in the regulation of IL-21R turnover and signaling, better define the effect of TGF β on Grail expression and to better characterized the functional consequences of the loss of Grail in CTLs.

-We would like to thank the reviewer for their feedback and critiques for the improvement of the manuscript.

2. I just want to point a couple of minor details the authors may want to address. Fig.2d shows increased infiltration not only of donor Grail-deficient CD8⁺ T cells bur also of host Grail-sufficient CTLs when OT-I Grail-/- CD8⁺ T cell s are transferred into tumor-bearing mice. This could be explained in the line suggested by one of the reviewers that maybe increased tumor death induced by the rail-deficient CD8⁺ T cells may lead to increased infiltration of not only donor but also host T cells. It would interesting if the authors could speculate in the discussion possible explanations for that effect.

-We would like to thank the reviewer for this feedback and have added a line in the results accordingly.

3. Finally, the legend of Suppl. Fig 1 indicates the experiments were performed twice but not how many mice were analyzed per experiment.

-We have updated the **Supplementary Figure 1** legend accordingly.

Reviewers' comments:

Reviewer #1 (Remarks to the Author):

Although the authors have addressed most of my previous concerns, I am still troubled by some of the flow cytometry data (specifically Fig. 4e and Fig 5E). The authors responded that they were using naïve T cells as the biological control as opposed to isotype controls, and I still think that it is imperative that the appropriate isotype control be used for each cell type and each stimulation condition, as activation can affect non-specific binding of antibodies. This is especially important for the IL-21R antibody, which has an extremely low level of staining.

Concerning the data in Figure 7 related to patient lymphomas, I continue to feel that these experiments are correlative and add little of substance to the paper unless the kind of data presented in the response letter are included.

Reviewer #3 (Remarks to the Author):

In Question 2, the authors have shown the effect of WT Tregs on GzmB/IFN γ expression in WT and Grail $^{-/-}$ CD8 T cells, but they have failed to provide the data for Grail $^{-/-}$ Treg-treated wild-type or Grail $^{-/-}$ CD8 T cells (new Figure 4a, histogram on the right). I do not understand why this has not been done. In any case, the results of this experiment change the conclusion/discussion of the paper, as it appears as though Grail $^{-/-}$ T regs play at least some role in tumour control.

Action: Please change the Discussion accordingly.

The requested experiment has been provided. I have to say that the tumor growth curve (new Supplementary Figure 2a) looks odd: I would expect that in Grail $^{-/-}$ mice, it would either be rejected or grow out of control. I still wonder whether this has anything to do with T regs.

Action: none requested

With respect to Question 4, I would like to refer the authors to Ln 154-156, where they have stated "We detected a statistically significant increase in mRNA levels of IFN- γ , granzyme B, TNF- α and perforin 1 in CD8+ TIL from Grail $^{-/-}$ mice (Supplementary Fig. 4c), further supporting that Grail controls the effector function of CTLs infiltrated into the tumors."

Action: The authors should make it clear that their data only shows that Grail $^{-/-}$ CD8 can express cytokines and effector molecules. They have never compared effector protein expression levels and, therefore, cannot conclude that those factors could play a role. The authors should examine proteins levels specifically, as mRNA levels may not be indicative of differences at the level of proteins, particularly with relatively modest changes in mRNA expression.

POINT-BY-POINT RESPONSE TO REVIEWERS' COMMENTS

We thank all reviewers for their critiques, comments and helpful advice. We have provided additional textual clarification as well as an additional control experiment demonstrating specific staining by the IL-21R antibody. We hope these edits will address all the critiques satisfactorily. All changes in the text are underlined.

Reviewers' comments

Reviewer #1:

1. *Although the authors have addressed most of my previous concerns, I am still troubled by some of the flow cytometry data (specifically Fig. 4e and Fig 5E). The authors responded that they were using naïve T cells as the biological control as opposed to isotype controls, and I still think that it is imperative that the appropriate isotype control be used for each cell type and each stimulation condition, as activation can affect non-specific binding of antibodies. This is especially important for the IL-21R antibody, which has an extremely low level of staining.*

- Per reviewer request, we have performed the requested isotype control staining to confirm the specificity of IL-21R antibodies and added this to the supplementary data (**Supplementary Fig. 9**).

2. *Concerning the data in Figure 7 related to patient lymphomas, I continue to feel that these experiments are correlative and add little of substance to the paper unless the kind of data presented in the response letter are included.*

- We regret that we are unable to add this data to the manuscript but have adjusted the language in the manuscript's results and discussion sections so as to not over interpret the current data.

Reviewer #3:

1. *In Question 2, the authors have shown the effect of WT Tregs on GzmB/IFN γ expression in WT and Grail $^{-/-}$ CD8 T cells, but they have failed to provide the data for Grail $^{-/-}$ Treg-treated wild-type or Grail $^{-/-}$ CD8 T cells (new Figure 4a, histogram on the right). I do not understand why this has not been done. In any case, the results of this experiment change the conclusion/discussion of the paper, as it appears as though Grail $^{-/-}$ T regs play at least some role in tumour control.*

-The Discussion has been changed accordingly to reviewer's advice.

2. *The requested experiment has been provided. I have to say that the tumor growth curve (new Supplementary Figure 2a) looks odd: I would expect that in Grail $^{-/-}$ mice, it would either be rejected or grow out of control. I still wonder whether this has anything to do with T regs.*

-We agree with the reviewer that this is a possibility and may be the direction of future study.

3. With respect to Question 4, I would like to refer the authors to Ln 154-156, where they have stated "We detected a statistically significant increase in mRNA levels of IFN- γ , granzyme B, TNF- α and perforin 1 in CD8⁺ TIL from Grail^{-/-} mice (Supplementary Fig. 4c), further supporting that Grail controls the effector function of CTLs infiltrated into the tumors."

- We agree with the reviewer that protein expression is the important in concluding the impact of cytotoxicity/effector function. As the focus of Grail's impact is on IFN γ and granzyme B expression, we assessed changes in these proteins by flow cytometry and ELISA throughout the manuscript. The protein data corresponding to **Supplementary Fig. 4c** is presented in **Fig. 1e**.